# Overcoming Common Flaws in the Evaluation of Selective Classification Systems

**Jeremias Traub**[1,2]   **Till J. Bungert**[1,2,6]   **Carsten T. Lüth**[1,2]   **Michael Baumgartner**[2,3,6]
**Klaus H. Maier-Hein**[2,3,5,6,7]   **Lena Maier-Hein**[2,4,6,7]   **Paul F. Jäger**[1,2]

[1]German Cancer Research Center (DKFZ) Heidelberg, Interactive Machine Learning Group, Germany
[2]Helmholtz Imaging, DKFZ Heidelberg, Germany
[3]DKFZ Heidelberg, Division of Medical Image Computing (MIC), Germany
[4]DKFZ Heidelberg, Division of Intelligent Medical Systems (IMSY), Germany
[5]Pattern Analysis and Learning Group, Department of Radiation Oncology,
Heidelberg University Hospital, 69120 Heidelberg, Germany
[6]Faculty of Mathematics and Computer Science, University of Heidelberg, Germany
[7]National Center for Tumor Diseases (NCT) Heidelberg
`jeremias.traub@dkfz-heidelberg.de`

## Abstract

Selective Classification, wherein models can reject low-confidence predictions, promises reliable translation of machine-learning based classification systems to real-world scenarios such as clinical diagnostics. While current evaluation of these systems typically assumes fixed working points based on pre-defined rejection thresholds, methodological progress requires benchmarking the general performance of systems akin to the AUROC in standard classification. In this work, we define 5 requirements for multi-threshold metrics in selective classification regarding task alignment, interpretability, and flexibility, and show how current approaches fail to meet them. We propose the Area under the Generalized Risk Coverage curve (AUGRC), which meets all requirements and can be directly interpreted as the average risk of undetected failures. We empirically demonstrate the relevance of AUGRC on a comprehensive benchmark spanning 6 data sets and 13 confidence scoring functions. We find that the proposed metric substantially changes metric rankings on 5 out of the 6 data sets.

## 1   Introduction

Selective Classification (SC) is increasingly recognized as a crucial component for the reliable deployment of machine learning-based classification systems in real-world scenarios, such as clinical diagnostics [Dvijotham et al., 2023, Leibig et al., 2022, Dembrower et al., 2020, Yala et al., 2019]. The core idea of SC is to equip models with the ability to reject low-confidence predictions, thereby enhancing the overall reliability and safety of the system [Geifman and El-Yaniv, 2017, Chow, 1957, El-Yaniv et al., 2010]. This is typically achieved through three key components: a classifier that makes the predictions, a confidence scoring function (CSF) that assesses the reliability of these predictions, and a rejection threshold $\tau$ that determines when to reject a prediction based on its confidence score.

In evaluating SC systems, two primary concepts are essential: risk and coverage. Risk expresses the classifier's error potential and is typically measured by its misclassification rate. Coverage, on the other hand, indicates the proportion of instances where the model makes a prediction rather than rejecting it. An effective SC system aims to minimize risk while maintaining high coverage, ensuring that the model provides accurate predictions for as many instances as possible.

Current evaluation of SC systems often focuses on fixed working points defined by pre-set rejection thresholds[Geifman and El-Yaniv, 2017, Liu et al., 2019, Geifman and El-Yaniv, 2019]. For instance, a common evaluation metric might be the selective risk at a given coverage of 70% [Geifman and El-Yaniv, 2019], which communicates the potential risk associated with a specific confidence score c and selection threshold $\tau$ to a patient or end-user. While this method is useful for communicating risk in specific instances, it does not provide a comprehensive evaluation of the system's overall performance. In standard classification, this is analogous to the need for the Area Under the Receiver Operating Characteristic (AUROC) curve rather than evaluating sensitivity at a specific specificity, which can be highly misleading when assessing general classifier capabilities [Maier-Hein et al., 2022]. The AUROC provides a holistic view of the classifier's performance across all possible thresholds, thereby driving methodological progress. Similarly, SC requires a multi-threshold metric that aggregates performance across all rejection thresholds to fully benchmark the system's capabilities.

Several current approaches attempt to address this need for multi-threshold metrics in SC. The Area Under the Risk Coverage curve (AURC) is the most prevalent of these Geifman et al. [2018]. However, we demonstrate in this work that AURC has significant limitations as it fails to adequately translate the risk from specific working points into a meaningful aggregated evaluation score. This shortcoming hampers the ability to benchmark and improve SC methodologies effectively.

In our work, we aim to bridge this gap by providing a robust evaluation framework for SC. Our contributions are as follows:

**Refined Task Formulation:** We are the first to provide a comprehensive SC evaluation framework, explicitly deriving meaningful task formulations for different evaluation and application scenarios such as working point versus multi-threshold evaluation.

**Formulation of Requirements:** We define five critical requirements for multi-threshold metrics in SC, focusing on task suitability, interpretability, and flexibility. We assess current multi-threshold metrics against our five requirements and demonstrate their shortcomings.

**Proposal of AUGRC:** We introduce the Area Under the Generalized Risk Coverage (AUGRC) curve, a new metric designed to overcome the flaws of current multi-threshold metrics for SC. AUGRC meets all five requirements, providing a comprehensive and interpretable measure of SC system performance.

**Empirical Validation:** We empirically demonstrate the relevance and effectiveness of AUGRC through a comprehensive benchmark spanning 6 datasets and 13 confidence scoring functions.

In summary, our work presents a significant advancement in the evaluation of SC systems, offering a more reliable and interpretable metric that can drive further methodological progress in the field.

## 2 Refined Task Formulation

Selective Classification (SC) systems consist of a classifier $m : \mathcal{X} \to \mathcal{Y}$, which outputs a prediction and a confidence scoring function (CSF) $g : \mathcal{X} \to \mathbb{R}$, which outputs a confidence score associated with the prediction. Assuming a supervised training setup, let $\{(x_i, y_i)\}_{i=1}^{N}$ be a dataset containing $N$ independent samples from the source distribution $P(X, Y)$ over $\mathcal{X} \times \mathcal{Y}$. Given a rejection threshold $\tau$, the model prediction is accepted only if the corresponding score is larger than $\tau$:

$$(m, g)(x) := \begin{cases} m(x), & \text{if } g(x) \geq \tau \\ \text{reject}, & \text{otherwise} \end{cases} \tag{1}$$

Given an error function $\ell : \mathcal{Y} \times \mathcal{Y} \to \mathbb{R}^+$, the overall risk of the classifier $m$ is given by $R(m) := \frac{1}{N} \sum_{i=1}^{N} \ell(m(x), y)$. The error function thereby contains the measure of classification performance suitable for the task at hand. Commonly, SC literature assumes a 0/1 error corresponding to the failure indicator variable $Y_f$ with $y_{f,i}(x_i, y_i, m) := \mathbb{I}[y_i \neq m(x_i)]$. In this case, the overall risk represents the probability of misclassification $P(Y_f = 1)$. By rejecting low-confidence predictions, the selective classifier can decrease the risk of misclassifications at the cost of letting the classifier make predictions on only a fraction of the dataset. This fraction is denoted as the coverage := $P(g(x) \geq \tau)$, with the respective empirical estimator $\frac{1}{N} \sum_{i=1}^{N} \mathbb{I}(g(x_i) \geq \tau)$. Evaluation of SC

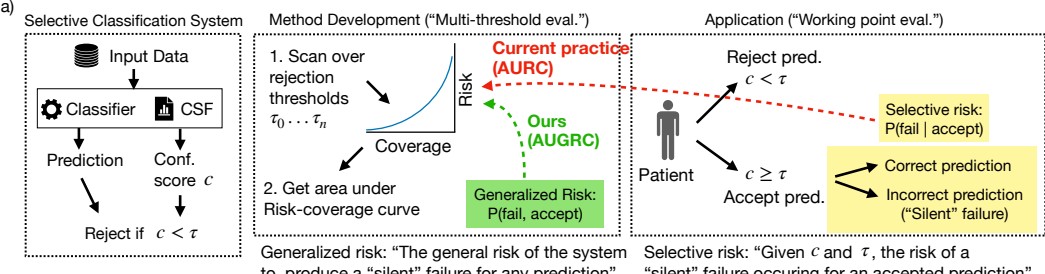

Figure 1: **The AUGRC metric based on Generalized Risk overcomes common flaws in current evaluation of Selective classification (SC). a)** Refined task definition for SC. Analogously to standard classification, we distinguish between holistic evaluation for method development and benchmarking using multi-threshold metrics versus evaluation of specific application scenarios at pre-determined working points. The current most prevalent multi-threshold metric in SC, AURC, is based on Selective Risk, a concept for working point evaluation that is not suitable for aggregation over rejection thresholds (red arrow). To fill this gap, we formulate the new concept of Generalized Risk and a corresponding metric, AUGRC (green arrow). **b)** We formalize our perspective on SC evaluation by identifying five key requirements for multi-threshold metrics and analyze how previous metrics fail to fulfill them. Abbreviations, CSF: Confidence Scoring Function, AU(G)RC: Area Under the (Generalized) Risk Coverage curve.

systems often focuses on preventing "silent", i.e. undetected failures for which both $y_{\mathrm{f},i} = 1$ and $g(x_i) \geq \tau$.

Deploying a SC system in practice requires the selection of a fixed rejection threshold $\tau$. In the following refined evaluation protocol, we distinguish between the well-established application-specific task formulation where SC models are evaluated at individual working points (Section 2.1) and SC method development where evaluation is independent of individual working points (Section 2.2).

## 2.1 Evaluating SC systems in applied settings

Current Selective Classification evaluation commonly reports the selective risk of the system at a given coverage ("risk@coverage"), which implies a pre-determined cutoff on the risk or coverage (e.g. "maximum risk of 5%") leading to a working point of the system defined by a rejection threshold $\tau$. The selective risk is defined as:

$$\text{Selective Risk}_{(m,g)}(\tau) \coloneqq \frac{\sum_{i=1}^{N} \ell(m(x_i), y_i) \cdot \mathbb{I}(g(x_i) \geq \tau)}{\sum_{i=1}^{N} \mathbb{I}(g(x_i) \geq \tau)} \tag{2}$$

This formulation only considers accepted predictions with $c > \tau$. For a binary failure error, this risk is an empirical estimator for the conditional probability $P(Y_{\mathrm{f}} = 1 | g(x) \geq \tau)$. Thus, this metric effectively communicates the risk of a "silent" failure for a specific prediction of classifier $m$ that has been accepted ($c > \tau$) at a pre-defined threshold $\tau$. This information may be useful in applied medical settings, e.g. to communicate the risk of misclassification for a patient p whose associated

prediction has been accepted by a given SC system ($c_p > \tau$). Aside from the selective risk, other performance measurements for working point selection include for example Classification quality and Rejection quality [Condessa et al., 2017].

## 2.2 Evaluating SC systems for method development and benchmarking

Evaluating the general performance of a Selective Classification system requires a multi-threshold metric for a more comprehensive view compared to a working point analysis based on a fixed singular rejection threshold, analogous to binary classification tasks, where the Area Under the Receiver Operating Characteristic (AUROC) is used to assess the entire system performance across multiple classifier thresholds. However, the working point analysis can not be simply translated to multi-threshold aggregation, as it is based on Selective Risk, which only considers the risk w.r.t accepted predictions, assuming that a specific selection has already occurred ($P(Y_\mathrm{f} = 1 \mid g(x) \geq \tau)$). This evaluation ignores rejected cases and thus contradicts the holistic assessment, where we are interested in the general risk for *any* prediction causing a silent failure independent of the future selection decision of individual predictions. This holistic assessment is reflected by the joint probability $P(Y_\mathrm{f} = 1, g(x) \geq \tau)$, which reflects the risk of silent failure for any prediction processed by the system including cases that are potentially rejected in the future.

To address the discrepancy between the need for holistic assessment versus the selective risk's focus on already selected cases, we formulate the generalized risk:

$$\text{Generalized Risk}(m, g)(\tau) := \frac{1}{N} \sum_{i=1}^{N} \ell(m(x_i), y_i) \cdot \mathbb{I}[g(x_i) \geq \tau]. \tag{3}$$

This metric reflects the joint probability of misclassification and acceptance by the confidence score threshold. By aggregating this risk over multiple rejection thresholds, we can evaluate the overall performance of the SC system.

## 2.3 Requirements for Selective Classification multi-threshold metrics

Based on the refined task definition given above, we can formulate concrete five requirements R1-R5 for multi-threshold metrics in Selective Classification:

**R1: Task Alignment.** The general goal in SC is to prevent silent failures either by preventing failures in the first place (via classifier performance), or by detecting the remaining failures (via CSF ranking quality). As argued in Jäger et al. [2023], it is crucial to jointly evaluate both aspects of an SC system, since the choice of CSF generally affects the underlying classifier. **R2: Monotonicity.** The metric should be monotonic w.r.t both evaluated factors stated in R1, i.e. improving on one of the two factors while keeping the other one fixed results in an improved metric value. Note that R2 does not make assumptions about how the two factors are combined, but it represents a minimum requirement for meaningful comparison and optimization of the metric. **R3: Ranking Interpretability.** Interpretability is a crucial component of a metric [Maier-Hein et al., 2022]. AUROC is the de facto standard ranking metric for binary classification tasks, providing an intuitive assessment of ranking quality which is proportional to the number of permutations needed for establishing an optimal ranking. This can also be interpreted as the "probability of a positive sample having a higher score than a negative one". Ideally, the SC metric should follow this intuitive assessment of rankings. **R4: CSF Flexibility.** The metric should be applicable to arbitrary choices of CSFs. This includes external CSFs, i.e. scores that are not based directly on the classifier output. **R5: Error Flexibility.** Current SC literature largely focuses on 0/1 error (i.e. $1 - \text{accuracy}$) in their risk computation. However, in many real-world scenarios, accuracy is not an adequate classification metric, such as in the presence of class imbalance and for pixel-level classification. Thus, the SC metric should be flexible w.r.t the choice of error function.

## 2.4 Current multi-threshold metrics in SC do not fulfill requirements R1-R5

**AURC:** Geifman et al. [2018] derive the AURC as the Area under the Selective Risk-Coverage curve. This metric is the most prevalent multi-threshold metric in SC [Geifman et al., 2018, Jäger et al., 2023, Bungert et al., 2023, Cheng et al., 2023, Zhu et al., 2023a, Varshney et al., 2020, Naushad and Voiculescu, 2024, Van Landeghem et al., 2024, 2023, Zhu et al., 2022, Xin et al.,

2021, Yoshikawa and Okazaki, 2023, Ding et al., 2020, Zhu et al., 2023b, Galil and El-Yaniv, 2021, Franc et al., 2023, Cen et al., 2023, Xia and Bouganis, 2022, Cattelan and Silva, 2023, Tran et al., 2022, Kim et al., 2023, Ashukha et al., 2020, Xia et al., 2024]. For the 0/1-error, AURC can be expressed through the following integral:

$$\text{AURC} = \int_0^1 P(Y_f = 1 | g(x) \geq \tau) \, dP(g(x) \geq \tau) \tag{4}$$

This integral effectively aggregates the Selective Risk (Equation 2) over all fractions of accepted predictions (i.e. coverage). However, as discussed in Section 2.2, the Selective Risk is not suitable for aggregation over rejection thresholds to holistically assess a SC system. This inadequacy effectively leads to an excessive over-weighting of high-confidence failures in AURC and in return to breaching the requirements of monotonicity (R2) and ranking interpretability (R3). We provide empirical evidence for these shortcomings on toy data (Section 3) and real-world data (Section 4.2). Further, current SC literature usually employs the 0/1 error function, even though the related Accuracy metric is well known to be an unsuitable classification performance measure for many applications. For example, Balanced Accuracy is used to address class imbalance and metrics such as the Dice Score are employed for segmentation. Moving beyond the 0/1 error yields a higher variance in distinct error values and may thus amplify the shortcoming of over-weighting high-confidence failures. As more sophisticated error functions being are an important direction of future SC research, it is crucial to ensure the compatibility of metrics in SC evaluation.

**e-AURC:** Geifman et al. [2018] further introduce the excess-AURC as the difference to the AURC of an optimal confidence scoring function (denoted as $\text{AURC}^*$)

$$\text{e-AURC} = \text{AURC} - \text{AURC}^* \tag{5}$$

It relies on the intuition that subtracting the optimal AURC eliminates the contribution of the overall classification performance and collapses it to a pure ranking metric [Geifman et al., 2018, Galil et al., 2023, Jäger et al., 2023]. However, several works demonstrated that this intuition does not hold and that the e-AURC is still sensitive to the overall classification performance [Galil et al., 2023, Cattelan and Silva, 2023]. We attribute this deviation from the intended behavior to the shortcomings of the underlying AURC, as we find the desired isolation of classifier performance in our improved metric formulation in Section 3. e-AURC further inherits the shortcomings of AURC w.r.t monotonicity (R2), ranking interpretability (R3), and error flexibility (R5).

**NLL / Brier Score:** Importantly, proper scoring rules such as the Negative-Log-Likelihood and the Brier Score are technically not multi-threshold metrics. Yet, we include them here as they also aim for a holistic performance assessment, i.e. assessment beyond individual working points, by evaluating a general meaningfulness of scores based on the softmax-output of the classifier [Ovadia et al., 2019]. Thereby, they jointly assess ranking and calibration of scores, which dilutes the focus on ranking quality in the context of SC [Jäger et al., 2023]. In our formulation, the calibration aspect in these metrics breaks the required monotonicity w.r.t SC evaluation (R2). Further, they are not applicable to CSFs beyond those that are directly based on the classifier output (R4). **AUROC_f:** The "failure version" of the standard AUROC assesses the correctness of predictions with a binary failure label [Jäger et al., 2023]. Based on the AUROC, it provides an intuitive ranking quality assessment. However, it ignores the underlying classifier performance (R1, R2), and is restricted to binary error functions (R5). **OC-AUC:** Kivlichan et al. [2021] introduce the Oracle-Model Collaborative AUC, where first a fixed threshold is applied on the confidence scores, above which error values are set to zero. Then, the $\text{AUROC}_f$ (or the Average Precision) are evaluated. This metric is also reported in [Dehghani et al., 2023, Tran et al., 2022], with a review fraction of 0.5%. OC-AUC is subject to the same pitfalls as $\text{AUROC}_f$ (R1, R2, R5). **AUROC-AURC:** Pugnana and Ruggieri [2023] propose the AUROC-AURC where the Selective Risk is defined as the classification (not failure) AUROC computed over the set of accepted predictions. However, it is only applicable to binary classification models with binary error functions (R5). Further, in employing Selective Risk it inherits the pitfalls described for AURC regarding monotonicity (R2) and ranking interpretability (R3). **NAURC:** Cattelan and Silva [2023] propose the Normalized-AURC, a min-max scaled version of the e-AURC, and claim that this modification eliminates its lack of interpretability. However, given the linear relationship with the AURC, it does not fulfill the requirements R2 and R3. **F1-AUC:** Malinin and Gales [2020], Malinin et al. [2021] introduce the notion of Error-Retention Curves, which corresponds to that of Generalized Risk Coverage Curves. However, in Malinin et al. [2021] the authors propose an F1-AUC metric which is only applicable to binary error functions (R5) and

breaks monotonicity (R2), as it decreases with increasing accuracy for accuracy values above $\approx 0.56$ (see Appendix A.1.3 for a detailed explanation.) **ARC:** Accuracy-Rejection-Curves [Nadeem et al., 2009, Ashukha et al., 2020, Condessa et al., 2017] and the associated AUC directly correspond to the AURC and are therefore subject to the same pitfalls (R2, R3).

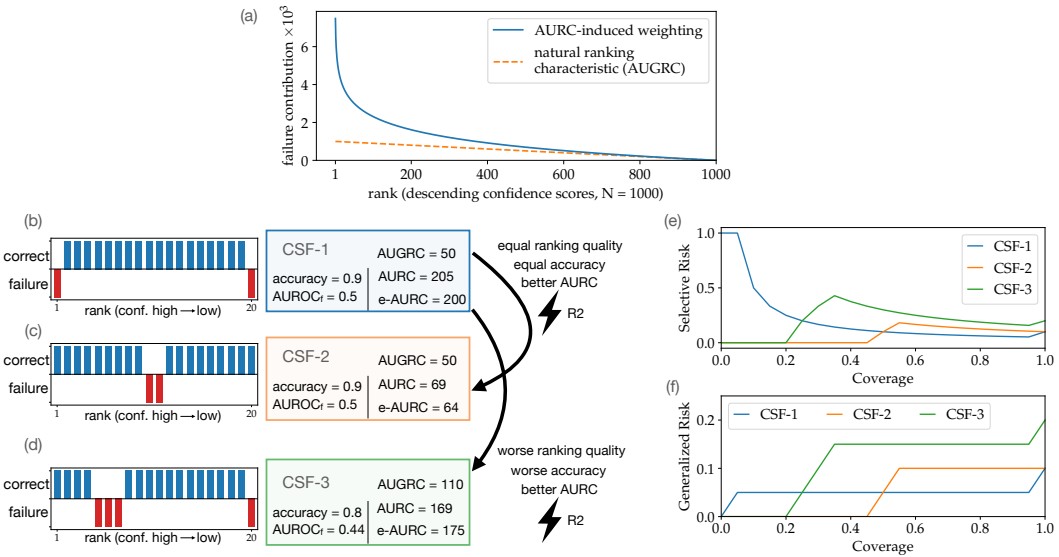

Figure 2: **The proposed AUGRC metric resolves shortcomings of AURC.** All figures are based on rankings of predictions according to descending associated confidence scores induced by a CSF. All AURC, e-AURC, and AUGRC values are scaled by $\times 1000$. **a)** shows the contribution of an individual failure case on the AURC and AUGRC metrics depending on its ranking position (for technical details, see Section A.1.1). While AUGRC reflects the intuitive behavior of weighing the failure cases proportional to their ranking position, the AURC puts excessive weight on high-confidence failures. **b-d)** Toy example of three CSFs ranking 20 predictions to show how AUGRC resolves the broken monotonicity requirement (R2) of AURC. Despite equal $\mathrm{AUROC_f}$ and equal $\mathrm{acc}$ in CSF-1 and CSF-2, the AURC improves. And AURC even improves in CSF-3, which features lower $\mathrm{AUROC_f}$ and lower $\mathrm{acc}$ compared to CSF-1. **e-f)** The corresponding risk-coverage curves reveal that the non-intuitive behavior of AURC is due to the excessive effect of the high-confidence failure of CSF-1 on the Selective Risk, which is resolved in the Generalized Risk.

## 3    Area under the Generalized Risk Coverage Curve

In Section 2.2, we illustrate that aggregating SC performance across working points requires to shift the perspective from the Selective Risk to the Generalized Risk (Equation 3) as an *holistic* assessment of the risk of silent failures for all predictions, before the rejection decision is made. We propose to evaluate SC methods via the Area under the Generalized Risk Coverage Curve (AUGRC). For the binary failure error it becomes an empirical estimator of the following expression:

$$\mathrm{AUGRC} = \int_0^1 P(Y_\mathrm{f} = 1,\, g(x) \geq \tau)\, \mathrm{d}P(g(x) \geq \tau) \tag{6}$$

This metric evaluates SC models in terms of the rate of silent failures averaged over working points and thus provides a practical measurement that is directly interpretable. It is bounded to the $[0, 1/2]$ interval, whereby lower AUGRC corresponds to better SC performance. The AUGRC is not subject to the shortcomings of the AURC, and we can derive a direct relationship to $\mathrm{AUROC_f}$ and $\mathrm{acc}$ (the derivation and visualizations are shown in Equation 8 and Figure 5):

$$\mathrm{AUGRC} = (1 - \mathrm{AUROC_f}) \cdot \mathrm{acc} \cdot (1 - \mathrm{acc}) + \frac{1}{2}(1 - \mathrm{acc})^2 \tag{7}$$

To illustrate, when drawing two random samples, it corresponds to the probability that either both are failure cases or that one is a failure and has a higher confidence score than the non-failure. This

is reflected by the second and first term of Equation 7, respectively. As an example, a CSF that outputs random scores yields $\mathrm{AUROC_f} = \frac{1}{2}$, and hence $\mathrm{AUGRC} = \frac{1}{2}(1 - \mathrm{acc})$. The AUGRC for an optimal CSF ($\mathrm{AUROC_f} = 1$) is given by the second term in Equation 7, hence subtracting the optimal SC performance yields a pure ranking measure re-scaled by the probability of finding a positive/negative pair. This overcomes the lack of interpretability of AURC and e-AURC (R3). Monotonicity (R2) is ensured since the partial gradients w.r.t. both $\mathrm{AUROC_f}$ and $\mathrm{acc}$ are always negative. Further, it can accommodate arbitrary classification metrics via the error function $\ell$ (R4) as well as arbitrary CSF (R5).

Figure 2a depicts the metric contribution of individual failure cases depending on their ranking position. This shows empirically how the excessive over-weighting of high-confidence failures in AURC is resolved by AUGRC, and how the intuitive ranking assessment of $\mathrm{AUROC_f}$ is established (see Section A.1.1 for a detailed derivation). We further showcase how AUGRC resolves the broken monotonicity requirement (R2) of AURC in Figures 2 b-d. Despite equal $\mathrm{AUROC_f}$ and equal $\mathrm{acc}$ in CSF-1 and CSF-2, the AURC improves. And AURC even improves in CSF-3, which features lower $\mathrm{AUROC_f}$ and lower $\mathrm{acc}$ compared to CSF-1. Figures 2 e-f depict the associated risk-coverage curves and reveal that the non-intuitive behavior of AURC is due to the excessive effect of the high-confidence failure of CSF-1 on the Selective Risk, which is resolved in the Generalized Risk. In Section 4.2 we demonstrate implications of AURC's shortcomings on real-world data.

## 4    Empirical Study

We conduct an empirical study based on the existing *FD-Shifts* benchmarking framework [Jäger et al., 2023], which compares various CSFs across a broad range of datasets. The focus of our study is not on the performance of individual methods (CSFs) but rather on the ranking of methods based on AUGRC evaluation as compared to AURC. We utilize the same experimental setup as in Jäger et al. [2023] with the addition of CSF scores based on temperature-scaled classifier logits. Extending the benchmark by recent SC methods, e.g. Feng et al. [2022], is an interesting direction of future work. A detailed overview of the datasets and methods used can be found in Appendix A.2. The code for reproducing our results and a PyTorch implementation of the AUGRC are available at: `https://github.com/IML-DKFZ/fd-shifts`.

### 4.1    Comparing Method Rankings of AUGRC and AURC

To study the relevance of AUGRC, we investigate the changes induced by AUGRC on rankings of CSFs compared to AURC method rankings.

**CSF Rankings with AUGRC substantially deviate from AURC rankings**. Figure 3 illustrates the method ranking differences for all i.i.d. test datasets, showing changes in ranks across all CSFs. Notably, AUGRC induces changes in the top-3 methods (out of 13) on 5 out of 6 datasets. To ensure that these ranking changes are not due to variability in the test data, we evaluate the metrics on 500 bootstrap samples from the test dataset and derive the compared method rankings based on the average rank across these samples ("rank-then-aggregate"). The size of each bootstrap sample corresponds to the size of the test dataset. Metric values for each bootstrap sample are averaged over 5 training initializations (2 for BREEDS and 10 for CAMELYON-17-Wilds). We analyze the robustness of the resulting method rankings for AURC and AUGRC separately, based on statistical significance. To that end, we perform pairwise CSF comparisons in form of a one-sided Wilcoxon signed-rank test [Wilcoxon, 1992] on the bootstrap samples. To control the family-wise error rate at a 5% significance level, we apply the Holm correction for multiple testing per metric and dataset, following [Wiesenfarth et al., 2021] (see Figure 7 for the results without correction for multiple testing). The resulting significance maps displayed in Figure 3 indicate stable method rankings for both AURC and AUGRC. This suggests that the observed ranking differences are induced by the conceptual difference of AUGRC compared to AURC and, as a consequence, that the shortcomings of current metrics and respective solutions discussed in Section 2, are not only conceptually sound, but also highly relevant in practice. For the results in Figure 3, we select the DeepGamblers reward parameter and whether to use dropout for both metrics separately on the validation dataset (details in Table 1 and Table 2).

**The shortcomings of AURC affect CSF comparison across datasets and distribution shifts**. The method rankings for all datasets and distribution shifts, based on the original test datasets, are shown

in Table 4, which also includes results for equal hyperparameters for AURC and AUGRC. Results for the bootstrap-based method ranking analysis for distribution shifts are displayed in Figure 6. The substantial differences in method rankings across all datasets and distribution shifts underline the relevance of AUGRC for SC evaluation. The complete AUGRC results on the *FD-Shifts* benchmark are shown in Table 3.

In few cases, as shown in Figure 3, we observe that a CSF A is ranked worse than a neighboring CSF B even though CSF A's ranks are significantly superior to those of CSF B based on the Wilcoxon statistic over the bootstrap samples. This discrepancy can occur if the rank variability in CSF A is larger than in CSF B. While this does not affect our conclusions regarding the relevance of the AUGRC, it indicates that selecting a single best CSF for application based solely on method rankings should be approached with caution. We recommend closer inspection of the individual CSF performance in such cases, although this is not the focus of our study.

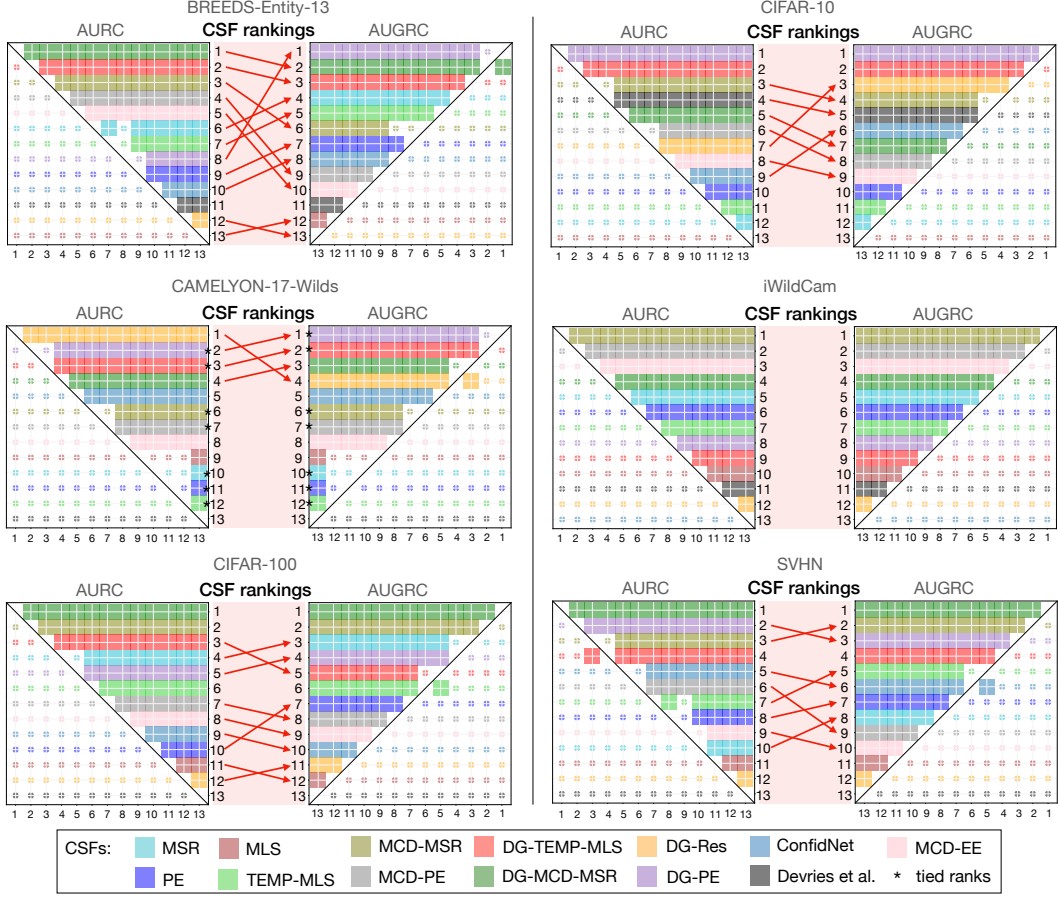

Figure 3: **Substantial differences in method rankings for AUGRC and AURC.** On 5 out of 6 datasets, the top-3 CSFs (out of 13 compared methods) change when employing the proposed AUGRC instead of AURC. This demonstrates the practical relevance of the AUGRC metric for Selective Classification evaluation. CSFs are color-coded and sorted from top (best) to bottom (worst) by average rank based on 500 bootstrap samples from the test dataset to ensure ranking stability. Ranking changes are reflected in changes in the color sequence and highlighted by red arrows. We assess the stability of the method rankings for each metric individually using one-sided Wilcoxon signed-rank tests based on the bootstrap samples at 5% significance level with adjustment for multiple testing according to Holm. Adjacent to each ranking, we present the resulting significance maps for the pairwise CSF comparisons. These maps can be interpreted as follows: At each grid position $(x, y)$, filled entries indicate that metric values of CSF $y$ are ranked significantly better than those from CSF $x$ (across bootstrap samples), cross-marks indicate no significant superiority. An ideal ranking exhibits only filled entries above the diagonal.

## 4.2 Showcasing Implications of AURC Shortcomings on Real-world Data

Figure 4 gives an in-depth look at how the conceptual shortcomings of AURC affect method assessment on real-world data. The example uses the confidence like reservation score of the DeepGamblers method (DG-Res) [Liu et al., 2019] and the Monte Carlo Dropout-based predictive entropy (MCD-PE) [Gal and Ghahramani, 2016] as CSFs on the CIFAR-10 test dataset. Despite DG-Res having higher classification performance and ranking quality than MCD-PE, the AURC erroneously favors MCD-PE over DG-Res. This violates the monotonicity requirement (R2). This can be attributed to the excessive contributions of only few high-confidence failures, which aligns with the theoretical findings on failure contributions shown in Figure 2 (R3). In this example, the high-confidence failures are associated with high label ambiguity or incorrect labeling, suggesting that the AURC may exacerbate the influence of label noise in practice.

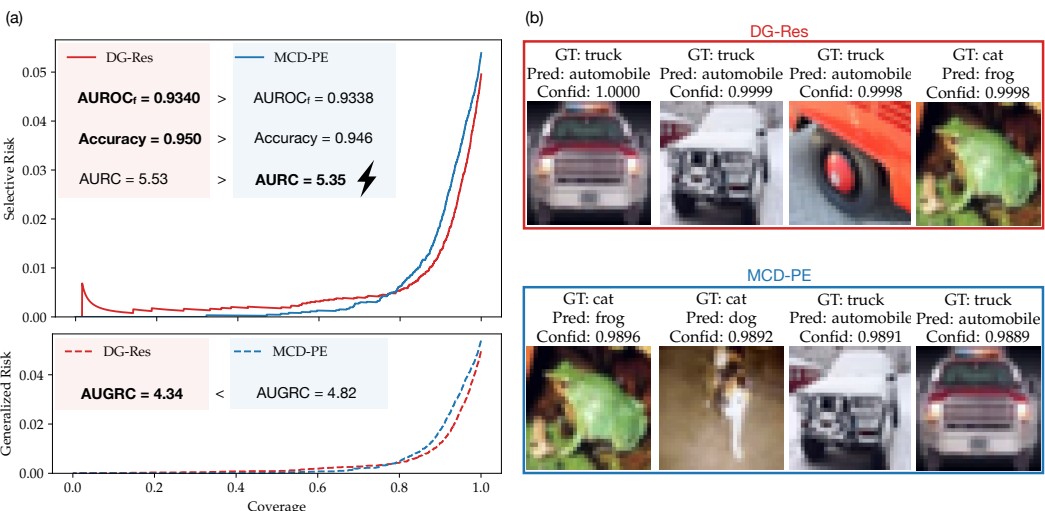

Figure 4: **The conceptual shortcomings of AURC affects method assessment in practice.** We illustrate the practical effects of excessive weight high-confidence failures in AURC by comparing the performance of two CSFs, DG-Res and MCD-PE, on the CIFAR-10 test dataset. (a) shows the coverage curves based on Selective Risk and Generalized Risk for both CSFs. The AURC violates the monotonicity requirement (R2) in practice, favoring MCD-PE despite a lower classification performance and ranking quality compared to DG-Res. (b) displays the images associated with the top-$k$ high-confidence failures. For DG-Res, the four failures correspond to the first four peaks in the Selective Risk curve, up to coverage $\approx 0.27$ (the total number of failures is 446). Only a few high-confidence failures significantly increase the AURC. For both CSFs, the images associated with high-confidence failures exhibit high label ambiguity or are incorrectly labeled, indicating that the AURC may amplify the influence of label noise in practice. AURC and AUGRC values are scaled by $\times 1000$.

## 5 Conclusion

Despite the increasing relevance of SC for reliable translation of machine learning systems to real-world application, we find that the current metrics have limitations in providing the comprehensive assessment needed to guide the methodological progress of SC systems.

In this work, we establish a systematic SC evaluation framework, thereby promoting the adoption of more comprehensive, interpretable, and task-aligned metrics for comparative benchmarking of SC systems.

We find that none of the existing multi-threshold metrics, particularly the AURC, meet the key requirements we identified for comprehensive SC evaluation, leading to deviations from intended and intuitive performance assessment behaviors. To address this, we introduce the AUGRC as a suitable metric for comprehensive SC method evaluation. Substantial differences in method rankings

between AURC and AUGRC, demonstrated through extensive empirical studies, highlight the importance of selecting the right SC metric. Thus, we propose the adoption of AUGRC for meaningful SC performance evaluation.

When evaluating specific applications for which certain risk or coverage intervals are known to be irrelevant, adaptations such as a partial AUGRC (analogous to the partial AUROC) may be considered. Further, investigating the relation between confidence ranking and calibration and on evaluating SC in settings where calibrated scores are of interest is an interesting direction of future work. Overall, our proposed evaluation framework provides a solid basis for future work in Selective Classification, including developing novel methods as well as analyzing the properties of individual methods.

## Acknowledgments and Disclosure of Funding

This work was partly funded by Helmholtz Imaging (HI), a platform of the Helmholtz Incubator on Information and Data Science. We thank Lukas Klein, Maximilian Zenk and Fabian Isensee for insightful discussions and feedback.

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

# A  Appendix

## A.1  Technical Details

### A.1.1  AURC Failure Contribution

Let $\{\tau_t\}_{t=1}^N$ provide a total order on the predictions (relaxing the assumption to a partial order leads to a more complicated but very similar formulation). Then, for a rejection threshold $\tau_t$, a failure case at rank $t^*$ contributes to the Selective Risk (Equation 2) by $\frac{1}{t}$ if $t^* \leq t$. Averaging over all thresholds yields a contribution through the failure at rank $t^*$ to the AURC of $\frac{1}{N} \sum_{t=t^*}^N \frac{1}{t}$. For the Generalized Risk, the contribution at $t^* \leq t$ is always $\frac{1}{N}$, resulting in an overall contribution of $\frac{N-t^*-1}{N^2}$ to the AUGRC. The failure contribution curves are displayed in Figure 2.

### A.1.2  Relationship between AUGRC and failure AUROC

Let $Y_{\mathrm{f}} = \mathbb{I}(m(x) \neq y)$ be a binary indicator variable for classification failures. Then, the Generalized Risk corresponds to the joint probability that a sample is accepted and wrongly classified. We can write the AUGRC as follows:

$$
\begin{aligned}
\mathrm{AUGRC} &= \int P(Y_{\mathrm{f}} = 1, g(x) \geq \tau)\, \mathrm{d}P(g(x) \geq \tau) \\
&= \int \left(1 - P(Y_{\mathrm{f}} = 0 | g(x) \geq \tau)\right) \cdot P(g(x) \geq \tau)\, \mathrm{d}P(g(x) \geq \tau) \\
&= \frac{1}{2} - \int P(g(x) \geq \tau | Y_{\mathrm{f}} = 0) P(Y_{\mathrm{f}} = 0)\, \mathrm{d}P(g(x) \geq \tau) \\
&\overset{(*)}{=} \frac{1}{2} - \mathrm{acc}\left[\frac{1}{2}\mathrm{acc} + (1 - \mathrm{acc}) \cdot \int P(g(x) \geq \tau | Y_{\mathrm{f}} = 0)\, \mathrm{d}P(g(x) \geq \tau | Y_{\mathrm{f}} = 1)\right] \\
&= \frac{1}{2} - \mathrm{acc}\left[\frac{1}{2}\mathrm{acc} + (1 - \mathrm{acc}) \cdot \mathrm{AUROC}_{\mathrm{f}}\right] \\
&= (1 - \mathrm{AUROC}_{\mathrm{f}}) \cdot \mathrm{acc}(1 - \mathrm{acc}) + \frac{1}{2}(1 - \mathrm{acc})^2
\end{aligned}
\tag{8}
$$

At $(*)$, we use that $P(g(x) \geq \tau) = P(g(x) \geq \tau | Y_{\mathrm{f}} = 0) \cdot \mathrm{acc} + P(g(x) \geq \tau | Y_{\mathrm{f}} = 1) \cdot (1 - \mathrm{acc})$. On choosing two random samples, the second term in the final equation represents the probability that both are failures and the first term represents the probability that one is a failure, the other is not, and that the failure has higher confidence score than the other.

The AUGRC is monotonic in $\mathrm{acc}$ and $\mathrm{AUROC}_{\mathrm{f}}$ as both partial derivatives are negative $\forall\, \mathrm{acc} \in (0, 1), \forall\, \mathrm{AUROC}_{\mathrm{f}} \in [0, 1]$:

$$
\frac{\partial \mathrm{AUGRC}}{\partial \mathrm{AUROC}_{\mathrm{f}}} = -\mathrm{acc} \cdot (1 - \mathrm{acc})
\tag{9}
$$

$$
\frac{\partial \mathrm{AUGRC}}{\partial \mathrm{acc}} = 2 \cdot \mathrm{AUROC}_{\mathrm{f}} \cdot \mathrm{acc} - \mathrm{acc} - \mathrm{AUROC}_{\mathrm{f}}
\tag{10}
$$

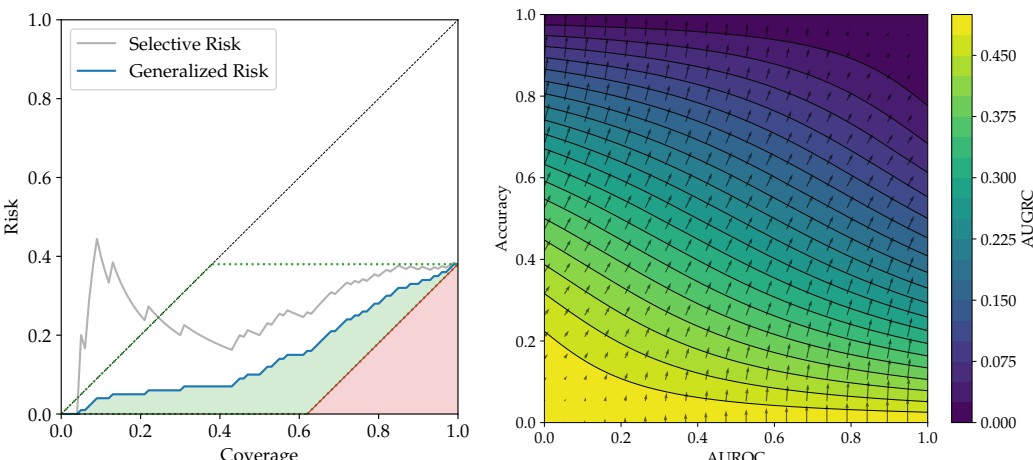

Figure 5: **Visualization of the relationship between AUGRC and AUROC_f.** (a) The Selective Risk curve can be transformed into the Generalized Risk curve via multiplication by the respective coverages. The resulting curve is monotonically increasing and bounded by the diagonal; decreasing Selective Risk corresponds to a plateau in Generalized Risk. The AUGRC corresponds to the AUGRC of an optimal CSF (shaded red) plus the re-scaled AUROC_f (shaded in green). The AUROC_f corresponds to the fraction of the area (parallelogram) enclosed by the green dashed line that lies above the Generalized Risk curve. (b) AUGRC (color-coded) and its negative gradients (arrows) in the Accuracy-AUROC_f space.

### A.1.3 F1-AUC does not fulfill R2

For the optimal CSF, all failure cases are attributed to lower confidence scores than correct predictions. Hence, the precision $P(Y_\mathrm{f} = 0|g(x) \geq \tau)$ is one for coverages up to $P(Y_\mathrm{f} = 0)$ and decreases as $P(Y_\mathrm{f} = 0)/P(g(x) \geq \tau)$ above. Following the task formulation as defined in Malinin et al. [2021], we derive the following analytical expression for the optimal F1-AUC*:

$$
\begin{aligned}
\mathrm{F1\text{-}AUC}^* &= \int_0^1 \mathrm{F1}(\tau)\, \mathrm{d}P(g(x) \geq \tau) \\
&= \int_0^{P(Y_\mathrm{f}=0)} \frac{2 \cdot P(g(x) \geq \tau)}{P(Y_\mathrm{f} = 0) + P(g(x) \geq \tau)}\, \mathrm{d}P(g(x) \geq \tau) \\
&\quad + \int_{P(Y_\mathrm{f}=0)}^1 \frac{2 \cdot P(Y_\mathrm{f} = 0)}{P(Y_\mathrm{f} = 0) + P(g(x) \geq \tau)}\, \mathrm{d}P(g(x) \geq \tau) \\
&= 2 \cdot \mathrm{acc} \cdot \left[1 + \ln\left(\frac{1}{4} \cdot (1 + \frac{1}{\mathrm{acc}})\right)\right]
\end{aligned}
\tag{11}
$$

F1-AUC* increases with increasing accuracy up to $P(Y_\mathrm{f} = 0) \approx 0.56$, then it decreases, favoring models with lower classification performance. Thus, the F1-AUC does not fulfill the monotonicity requirement (R2).

## A.2 Experiment Setup

### A.2.1 Datasets and Methods

We compare the following CSFs: From the classifier's logits we compute the Maximum Softmax Response (MSR), Maximum Logit Score (MLS), Predictive Entropy (PE), and the MLS based on temperature-scaled logits, for which a scalar temperature parameter is tuned based on the validation set [Guo et al., 2017]. Three predictive uncertainty measures are based on Monte Carlo Dropout [Gal and Ghahramani]: mean softmax (MCD-MSR), predictive entropy (MCD-PE), and mutual information (MCD-MI). We additionally include the DeepGamblers method [Liu et al.], which learns

a confidence like reservation score (DG-Res), ConfidNet Corbière et al. [2019], which is trained as an extension to the classifier, and the work of DeVries and Taylor [2018].

We evaluate SC methods on the *FD-Shifts* benchmark [Jäger et al., 2023], which considers a broad range of datasets and failure sources through various distribution shifts: SVHN [Netzer et al., 2011], CIFAR-10, and CIFAR-100 [Krizhevsky] are evaluated on semantic and non-semantic new-class shifts in a rotating fashion including Tiny ImageNet [Le and Yang, 2015]. Additionally, we consider sub-class shifts on iWildCam [Koh et al., 2021], BREEDS-Enity-13 [Santurkar et al., 2020], CAMELYON-17-Wilds [Koh et al., 2021], and on CIFAR-100 (based on super-classes) as well as corruption shifts based on Hendrycks and Dietterich [2019] on CIFAR-100. The data preparation and splitting are done as described in Jäger et al. [2023]. For the corruption shifts, we reduce the test split size to 75000 (subsampled within each corruption type and intensity level).

The following classifier architecture are used in the benchmark: small convolutional network for SVHN, VGG-13 [Simonyan and Zisserman, 2014] on CIFAR-10/100, ResNet-50 [He et al., 2016] on the other datasets.

If the distribution shift is not mentioned explicitly, we evaluate on the respective i.i.d. test datasets.

Our method ranking study focuses on the evaluation of CSF performance based on the existing *FD-Shifts* benchmark, hence we required no GPU's for the analysis in Section. 4. As both AURC and AUGRC can be computed efficiently, (CPU) evaluation time for a single test set is less than a minute; evaluation on 500 bootstrap samples on a single CPU core take around 3 hours.

### A.2.2 Hyperparameters and Model Selection

The experiments are based on the same hyperparameters as reported in Table 4 in Jäger et al. [2023]. Based on the performance on the validation set, we choose the DeepGambler reward hyperparameter and whether to use dropout (for the non-MCD-based CSFs). For the former, we select from $[2.2, 3, 6, 10]$ on Wilds-Camelyon-17, CIFAR-10, and SVHN, from $[2.2, 3, 6, 10, 15]$ on iWildCam and BREEDS-Entity-13, and from $[2.2, 3, 6, 10, 12, 15, 20]$ on CIFAR-100. When performing model selection based on the AURC metric, we obtain the same configurations as reported in Jäger et al. [2023]. When performing model selection based on the AUGRC metric, we obtain the parameters as reported in Tab. 1 and Tab. 2. For temperature scaling, we optimize the NLL on the validation set using the L-BFGS algorithm with $\mathrm{lr} = 0.01$.

Table 1: Selected DeepGambler reward hyperparameter based on the AUGRC on the validation set for all confidence scoring functions trained with the DeepGamblers objective.

| Method | iWildCam | BREEDS-Enity-13 | Wilds-Camelyon-17 | CIFAR-100 | CIFAR-10 | SVHN |
|---|---|---|---|---|---|---|
| DG-MCD-MSR | 15 | 15 | 10 | 20 | 3 | 3 |
| DG-PE | 15 | 15 | 2.2 | 10 | 10 | 3 |
| DG-Res | 6 | 2.2 | 6 | 12 | 2.2 | 2.2 |
| DG-TEMP-MLS | 15 | 15 | 2.2 | 15 | 10 | 3 |

Table 2: Whether or not dropout-training has been selected based on the AUGRC on the validation set. This selection is only done for deterministic confidence scoring methods (no MCD). "1" denotes dropout training and "0" denotes training without dropout.

| Method | iWildCam | BREEDS-Enity-13 | Wilds-Camelyon-17 | CIFAR-100 | CIFAR-10 | SVHN |
|---|---|---|---|---|---|---|
| MSR | 0 | 0 | 1 | 0 | 1 | 1 |
| MLS | 0 | 0 | 1 | 1 | 1 | 1 |
| PE | 0 | 0 | 1 | 0 | 1 | 1 |
| ConfidNet | 1 | 0 | 1 | 1 | 1 | 1 |
| DG-Res | 0 | 1 | 0 | 0 | 0 | 1 |
| Devries et al. | 1 | 0 | 1 | 0 | 0 | 1 |
| TEMP-MLS | 0 | 0 | 1 | 0 | 1 | 1 |
| DG-PE | 1 | 0 | 0 | 0 | 0 | 1 |
| DG-TEMP-MLS | 1 | 0 | 0 | 0 | 0 | 1 |

## A.3 AUGRC Computation Time

The main bottleneck of the AUGRC computation is the sort operation on the confidence scores. This is the same as for the AURC and there is no computational overhead compared to the AURC. For a small benchmark of the computational time, we evaluate the AURC, AUGRC, and $\text{AUROC}_f$ on 1000 random scores and failure labels. We obtain the following results (averaged over 5000 cases): $564\mu s \pm 2\mu s$ (AURC), $562\mu s \pm 2\mu s$ (AUGRC), $730\mu s \pm 13\mu s$ ($\text{AUROC}_f$).

## A.4 Additional Results

Table 3 shows the AUGRC results for the 13 compared CSFs across all datasets and distribution shifts. While the MSR baseline is not consistently outperformed across all experiments by any of the compared CSFs, MCD improves MSR scores in most settings. Temperature scaling does not exhibit consistent improvement of MLS scores.

Table 3: **FD-Shifts benchmark results measured as AUGRC $\times 10^3$ (score range: [0, 500], lower is better ↓).** The color heatmap is normalized per column, whereby whiter colors depict better scores. "cor" is the average over 5 intensity levels of image corruption shifts. AUGRC scores are averaged over 10 runs on CAMELYON-17-Wilds, over 2 runs on BREEDS, and over 5 runs on all other datasets. Abbreviations: ncs: new-class shift (s for semantic, ns for non-semantic), iid: independent and identically distributed, sub: sub-class shift, cor: image corruptions, c10/100: CIFAR-10/100, ti: TinyImagenet.

| study | iWildCam iid | iWildCam sub | BREEDS iid | BREEDS sub | CAMELYON iid | CAMELYON sub | CIFAR-100 iid | CIFAR-100 sub | CIFAR-100 cor | CIFAR-100 s-ncs c10 | CIFAR-100 ns-ncs svhn | CIFAR-100 ns-ncs ti | CIFAR-10 iid | CIFAR-10 cor | CIFAR-10 s-ncs c100 | CIFAR-10 ns-ncs svhn | CIFAR-10 ns-ncs ti | SVHN iid | SVHN ns-ncs c10 | SVHN ns-ncs c100 | SVHN ns-ncs ti |
|---|---|---|---|---|---|---|---|---|---|---|---|---|---|---|---|---|---|---|---|---|---|
| MSR | 48.0 | 55.0 | 6.86 | 117 | 8.13 | 91.3 | 54.6 | 150 | 187 | 210 | 334 | 203 | 4.93 | 67.9 | 166 | 291 | 154 | 3.44 | 48.5 | 48.2 | 48.6 |
| MLS | 60.3 | 62.4 | 9.28 | 121 | 8.13 | 91.3 | 65.6 | 155 | 200 | 213 | 337 | 195 | 5.56 | 68.4 | 163 | 288 | 150 | 3.97 | 46.5 | 46.1 | 46.6 |
| PE | 48.2 | 55.2 | 6.87 | 116 | 8.13 | 91.3 | 56.4 | 150 | 188 | 210 | 333 | 202 | 4.91 | 67.2 | 165 | 290 | 152 | 3.44 | 47.9 | 47.5 | 48.0 |
| MCD-MSR | 42.8 | 54.3 | 6.89 | 116 | 5.64 | 95.1 | 53.1 | 146 | 176 | 210 | 336 | 202 | 4.56 | 60.4 | 166 | 294 | 156 | 3.40 | 47.3 | 47.4 | 47.9 |
| MCD-PE | 43.4 | 55.1 | 6.98 | 116 | 5.64 | 95.1 | 56.6 | 146 | 179 | 210 | 335 | 198 | 4.73 | 60.6 | 164 | 293 | 153 | 3.47 | 46.3 | 46.4 | 46.9 |
| MCD-MI | 43.7 | 58.1 | 7.28 | 118 | 6.13 | 101 | 59.4 | 148 | 182 | 211 | 333 | 197 | 4.99 | 64.9 | 167 | 302 | 161 | 3.55 | 47.0 | 47.7 | 48.2 |
| ConfidNet | 82.0 | 91.2 | 6.89 | 117 | 4.36 | 86.1 | 57.5 | 154 | 192 | 214 | 340 | 200 | 4.70 | 65.4 | 165 | 291 | 153 | 3.43 | 48.6 | 48.4 | 48.8 |
| DG-MCD-MSR | 43.9 | 59.6 | 6.31 | 112 | 3.47 | 149 | 52.5 | 145 | 175 | 210 | 336 | 203 | 4.75 | 60.8 | 168 | 296 | 159 | 3.38 | 47.6 | 47.7 | 48.0 |
| DG-Res | 66.5 | 63.8 | 9.39 | 122 | 3.46 | 124 | 64.3 | 230 | 194 | 215 | 330 | 197 | 4.40 | 63.8 | 167 | 287 | 151 | 4.09 | 46.8 | 46.2 | 46.6 |
| Devries et al. | 61.7 | 74.7 | 8.24 | 120 | 24.3 | 145 | 69.6 | 150 | 214 | 222 | 342 | 211 | 4.57 | 70.8 | 166 | 287 | 152 | 5.56 | 48.3 | 48.5 | 49.9 |
| TEMP-MLS | 48.2 | 55.1 | 6.86 | 116 | 8.13 | 91.3 | 54.8 | 150 | 187 | 210 | 333 | 203 | 4.92 | 67.6 | 166 | 291 | 153 | 3.44 | 48.3 | 47.9 | 48.4 |
| DG-PE | 52.0 | 69.7 | 6.33 | 114 | 3.46 | 123 | 55.2 | 151 | 186 | 210 | 329 | 197 | 4.09 | 62.9 | 167 | 288 | 154 | 3.41 | 48.9 | 48.4 | 48.7 |
| DG-TEMP-MLS | 52.2 | 69.9 | 6.35 | 114 | 3.46 | 123 | 55.0 | 222 | 184 | 211 | 331 | 199 | 4.09 | 63.1 | 167 | 288 | 155 | 3.41 | 49.0 | 48.5 | 48.8 |

Table 4: **Comparing Rankings of AURC → α versus AUGRC → β.** Differences in the method rankings between AURC and AUGRC demonstrate the relevance of the pitfalls of the AURC discussed in Section 2.4. Upper half: Model selection for dropout and DG hyperparameter was done for both metrics separately. Lower half: Model selection was done based on AUGRC for both α and β. The selected hyperparameters are reported in Appendix A.2.2. The color heatmap is normalized per column, whereby whiter colors depict better scores. "cor" is the average over 5 intensity levels of image corruption shifts. AUGRC scores are averaged over 10 runs on CAMELYON-17-Wilds, over 2 runs on BREEDS, and over 5 runs on all other datasets. Abbreviations: ncs: new-class shift (s for semantic, ns for non-semantic), iid: independent and identically distributed, sub: sub-class shift, cor: image corruptions, c10/100: CIFAR-10/100, ti: TinyImagenet.

| | iWildCam | | | | BREEDS | | | | CAMELYON | | | | CIFAR-100 | | | | | | | | | | | | CIFAR-10 | | | | | | | | | | SVHN | | | | | | |
|---|---|---|---|---|---|---|---|---|---|---|---|---|---|---|---|---|---|---|---|---|---|---|---|---|---|---|---|---|---|---|---|---|---|---|---|---|---|---|---|---|---|---|
| study | iid | | sub | | iid | | sub | | iid | | sub | | iid | | sub | | cor | | s-ncs | | ns-ncs | | | | iid | | cor | | s-ncs | | ns-ncs | | | | iid | | ns-ncs | | | | | |
| ncs-data set | | | | | | | | | | | | | | | | | | | c10 | | svhn | | ti | | | | | | c100 | | svhn | | ti | | | | c10 | | c100 | | ti | |
| metric | α | β | α | β | α | β | α | β | α | β | α | β | α | β | α | β | α | β | α | β | α | β | α | β | α | β | α | β | α | β | α | β | α | β | α | β | α | β | α | β | α | β |
| **optimized both (w.r.t. dropout and DG reward parameter)** | | | | | | | | | | | | | | | | | | | | | | | | | | | | | | | | | | | | | | | | | | |
| MSR | 5 | 5 | 6 | 2 | 6 | 4 | 7 | 9 | 9 | 9 | 2 | 2 | 3 | 3 | 5 | 5 | 7 | 7 | 3 | 1 | 3 | 6 | 11 | 10 | 12 | 12 | 11 | 11 | 5 | 6 | 9 | 7 | 8 | 9 | 8 | 6 | 10 | 10 | 9 | 9 | 9 | 9 |
| MLS | 10 | 10 | 10 | 8 | 13 | 12 | 13 | 12 | 9 | 9 | 2 | 2 | 11 | 12 | 7 | 11 | 12 | 12 | 9 | 10 | 7 | 11 | 1 | 1 | 13 | 13 | 12 | 12 | 1 | 1 | 2 | 3 | 1 | 1 | 11 | 11 | 3 | 3 | 1 | 1 | 1 | 1 |
| PE | 6 | 6 | 7 | 5 | 8 | 6 | 6 | 4 | 9 | 9 | 2 | 2 | 10 | 7 | 9 | 5 | 11 | 9 | 10 | 1 | 12 | 4 | 6 | 8 | 10 | 10 | 9 | 9 | 4 | 4 | 6 | 6 | 2 | 3 | 8 | 6 | 7 | 7 | 6 | 6 | 6 | 6 |
| MCD-MSR | 1 | 1 | 1 | 1 | 3 | 7 | 3 | 4 | 6 | 6 | 6 | 7 | 2 | 2 | 2 | 2 | 2 | 2 | 1 | 1 | 9 | 8 | 9 | 8 | 3 | 4 | 2 | 2 | 5 | 6 | 12 | 12 | 10 | 12 | 4 | 2 | 5 | 5 | 5 | 5 | 5 | 5 |
| MCD-PE | 2 | 2 | 2 | 3 | 4 | 9 | 3 | 4 | 6 | 6 | 6 | 7 | 7 | 8 | 2 | 2 | 3 | 3 | 1 | 1 | 6 | 7 | 3 | 4 | 5 | 7 | 3 | 3 | 3 | 2 | 11 | 11 | 8 | 6 | 6 | 9 | 1 | 1 | 4 | 4 | 4 | 4 |
| MCD-EE | 3 | 3 | 3 | 6 | 5 | 10 | 3 | 4 | 8 | 6 | 6 | 6 | 8 | 9 | 2 | 2 | 4 | 4 | 6 | 8 | 7 | 8 | 6 | 6 | 7 | 9 | 1 | 1 | 2 | 2 | 9 | 7 | 2 | 3 | 8 | 10 | 2 | 2 | 3 | 3 | 3 | 3 |
| ConfidNet | 13 | 13 | 13 | 13 | 10 | 7 | 9 | 9 | 5 | 5 | 1 | 1 | 9 | 10 | 7 | 10 | 9 | 10 | 11 | 11 | 11 | 12 | 4 | 6 | 9 | 6 | 8 | 8 | 9 | 4 | 6 | 7 | 6 | 6 | 5 | 5 | 11 | 11 | 10 | 10 | 10 | 11 |
| DG-MCD-MSR | 4 | 4 | 9 | 7 | 1 | 1 | 1 | 1 | 4 | 4 | 12 | 13 | 1 | 1 | 1 | 1 | 1 | 1 | 3 | 1 | 10 | 8 | 11 | 10 | 6 | 8 | 4 | 4 | 10 | 13 | 13 | 13 | 13 | 13 | 1 | 1 | 6 | 6 | 7 | 7 | 7 | 6 |
| DG-Res | 12 | 12 | 11 | 9 | 12 | 13 | 12 | 13 | 1 | 1 | 9 | 11 | 12 | 11 | 13 | 13 | 10 | 11 | 12 | 12 | 1 | 2 | 2 | 2 | 8 | 3 | 7 | 7 | 13 | 10 | 3 | 1 | 5 | 2 | 12 | 12 | 4 | 4 | 2 | 2 | 2 | 1 |
| Devries et al. | 11 | 11 | 12 | 12 | 11 | 11 | 11 | 11 | 13 | 13 | 13 | 12 | 13 | 13 | 10 | 5 | 13 | 13 | 13 | 13 | 13 | 13 | 13 | 13 | 4 | 5 | 13 | 13 | 5 | 6 | 1 | 1 | 2 | 3 | 13 | 13 | 8 | 8 | 13 | 12 | 13 | 13 |
| TEMP-MLS | 7 | 6 | 8 | 3 | 8 | 4 | 7 | 4 | 9 | 9 | 2 | 2 | 5 | 4 | 5 | 5 | 8 | 7 | 3 | 1 | 3 | 4 | 10 | 10 | 11 | 11 | 10 | 10 | 5 | 6 | 8 | 7 | 6 | 6 | 7 | 6 | 8 | 8 | 8 | 8 | 8 | 8 |
| DG-PE | 8 | 8 | 4 | 10 | 7 | 2 | 10 | 2 | 1 | 1 | 10 | 9 | 6 | 6 | 11 | 9 | 6 | 6 | 6 | 1 | 2 | 1 | 5 | 2 | 1 | 1 | 5 | 5 | 11 | 10 | 4 | 3 | 10 | 9 | 2 | 3 | 12 | 12 | 10 | 10 | 10 | 10 |
| DG-TEMP-MLS | 8 | 9 | 4 | 11 | 3 | 2 | 2 | 2 | 1 | 1 | 10 | 9 | 4 | 5 | 12 | 12 | 5 | 5 | 8 | 8 | 3 | 3 | 6 | 5 | 1 | 1 | 6 | 6 | 11 | 10 | 5 | 3 | 12 | 11 | 3 | 3 | 13 | 13 | 12 | 12 | 12 | 11 |
| **optimized AUGRC (w.r.t. dropout and DG reward parameter)** | | | | | | | | | | | | | | | | | | | | | | | | | | | | | | | | | | | | | | | | | | |
| MSR | 5 | 5 | 4 | 2 | 7 | 4 | 8 | 9 | 9 | 9 | 2 | 2 | 3 | 3 | 7 | 5 | 7 | 7 | 4 | 1 | 5 | 6 | 11 | 10 | 12 | 12 | 11 | 11 | 5 | 6 | 9 | 7 | 8 | 9 | 8 | 6 | 10 | 10 | 9 | 9 | 9 | 9 |
| MLS | 10 | 10 | 8 | 8 | 13 | 12 | 13 | 12 | 9 | 9 | 2 | 2 | 11 | 12 | 10 | 11 | 12 | 12 | 10 | 10 | 8 | 11 | 1 | 1 | 13 | 13 | 12 | 12 | 1 | 1 | 2 | 3 | 1 | 1 | 11 | 11 | 3 | 3 | 1 | 1 | 1 | 1 |
| PE | 6 | 6 | 5 | 5 | 8 | 6 | 7 | 4 | 9 | 9 | 2 | 2 | 9 | 7 | 6 | 5 | 9 | 9 | 1 | 1 | 4 | 4 | 6 | 8 | 10 | 10 | 9 | 9 | 4 | 4 | 6 | 6 | 2 | 3 | 6 | 7 | 7 | 7 | 6 | 6 | 6 | 6 |
| MCD-MSR | 1 | 1 | 1 | 1 | 4 | 7 | 4 | 4 | 6 | 6 | 6 | 7 | 2 | 2 | 2 | 2 | 2 | 2 | 1 | 1 | 10 | 8 | 8 | 8 | 3 | 4 | 2 | 2 | 5 | 6 | 12 | 12 | 10 | 12 | 4 | 2 | 5 | 5 | 5 | 5 | 5 | 5 |
| MCD-PE | 2 | 2 | 2 | 3 | 5 | 9 | 4 | 4 | 6 | 6 | 6 | 7 | 7 | 8 | 2 | 2 | 3 | 3 | 1 | 1 | 7 | 7 | 4 | 4 | 5 | 7 | 3 | 3 | 3 | 2 | 11 | 11 | 8 | 6 | 6 | 9 | 1 | 1 | 4 | 4 | 4 | 4 |
| MCD-EE | 3 | 3 | 3 | 6 | 6 | 10 | 4 | 4 | 8 | 8 | 6 | 6 | 8 | 9 | 2 | 2 | 4 | 4 | 7 | 8 | 8 | 8 | 7 | 6 | 7 | 9 | 1 | 1 | 2 | 2 | 9 | 7 | 2 | 3 | 8 | 10 | 2 | 2 | 3 | 3 | 3 | 3 |
| ConfidNet | 13 | 13 | 13 | 13 | 10 | 7 | 10 | 9 | 5 | 5 | 1 | 1 | 10 | 10 | 10 | 10 | 10 | 10 | 11 | 11 | 12 | 12 | 5 | 6 | 9 | 6 | 8 | 8 | 9 | 4 | 6 | 7 | 6 | 6 | 5 | 5 | 11 | 11 | 10 | 10 | 10 | 11 |
| DG-MCD-MSR | 4 | 4 | 7 | 7 | 1 | 1 | 1 | 1 | 4 | 4 | 12 | 13 | 1 | 1 | 1 | 1 | 1 | 1 | 4 | 1 | 11 | 8 | 11 | 10 | 6 | 8 | 4 | 4 | 10 | 13 | 13 | 13 | 13 | 13 | 1 | 1 | 6 | 6 | 7 | 7 | 7 | 6 |
| DG-Res | 12 | 12 | 9 | 9 | 12 | 13 | 12 | 13 | 1 | 1 | 9 | 11 | 12 | 11 | 13 | 13 | 11 | 11 | 12 | 12 | 1 | 2 | 2 | 2 | 8 | 3 | 7 | 7 | 13 | 10 | 3 | 1 | 5 | 2 | 12 | 12 | 4 | 4 | 2 | 2 | 2 | 1 |
| Devries et al. | 11 | 11 | 12 | 12 | 11 | 11 | 11 | 11 | 13 | 13 | 13 | 12 | 13 | 13 | 5 | 5 | 13 | 13 | 13 | 13 | 13 | 13 | 13 | 13 | 4 | 5 | 13 | 13 | 5 | 6 | 1 | 1 | 2 | 3 | 13 | 13 | 8 | 8 | 13 | 12 | 13 | 13 |
| TEMP-MLS | 7 | 6 | 6 | 3 | 8 | 4 | 8 | 4 | 9 | 9 | 2 | 2 | 4 | 4 | 7 | 5 | 8 | 7 | 4 | 1 | 5 | 4 | 10 | 10 | 11 | 11 | 10 | 10 | 5 | 6 | 8 | 7 | 6 | 6 | 7 | 6 | 8 | 8 | 8 | 8 | 8 | 8 |
| DG-PE | 8 | 8 | 10 | 10 | 1 | 2 | 2 | 2 | 1 | 1 | 10 | 9 | 5 | 6 | 9 | 9 | 6 | 6 | 7 | 1 | 1 | 1 | 3 | 2 | 1 | 1 | 5 | 5 | 11 | 10 | 4 | 3 | 10 | 9 | 2 | 3 | 12 | 12 | 10 | 10 | 10 | 10 |
| DG-TEMP-MLS | 9 | 9 | 11 | 11 | 3 | 3 | 2 | 2 | 1 | 1 | 10 | 9 | 6 | 5 | 12 | 12 | 5 | 5 | 9 | 8 | 3 | 3 | 5 | 5 | 1 | 1 | 6 | 6 | 11 | 10 | 5 | 3 | 12 | 11 | 3 | 3 | 13 | 13 | 12 | 12 | 12 | 11 |

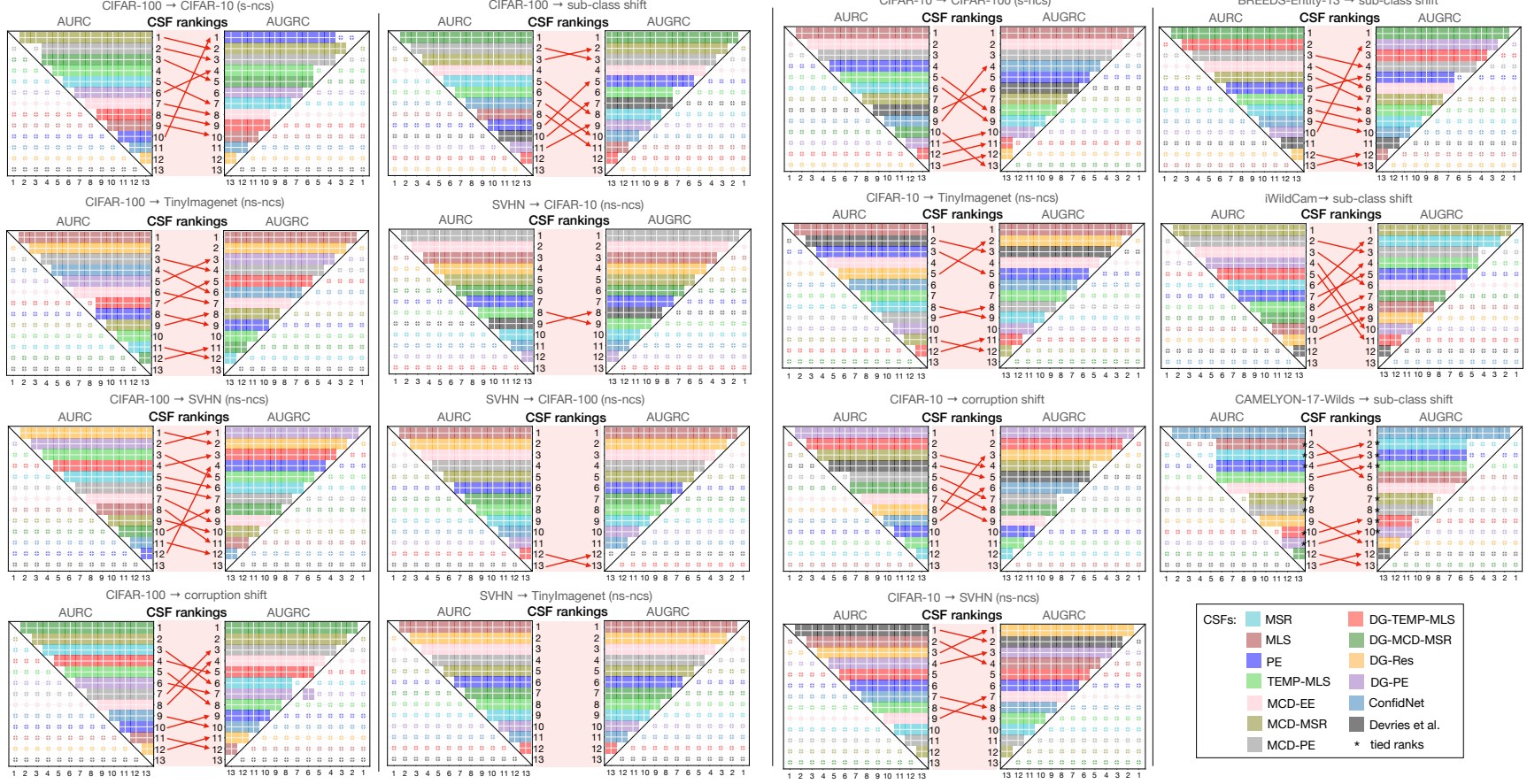

Figure 6: **Method ranking analysis for distribution shifts.** This figure displays the results for the method ranking analysis for AURC and AUGRC for evaluation under distribution shifts, analogous to Figure 3. The subfigures are titled as follows: "training dataset" → "evaluation dataset". Details on the distribution shifts are given in Section A.2.1. CSFs are color-coded and sorted from top (best) to bottom (worst) by average rank based on 500 bootstrap samples from the test dataset to ensure ranking stability. Ranking changes are reflected in changes in the color sequence and highlighted by red arrows. We assess the stability of the method rankings for each metric individually using one-sided Wilcoxon signed-rank tests based on the bootstrap samples at $5\%$ significance level with adjustment for multiple testing according to Holm. Adjacent to each ranking, we present the resulting significance maps for the pairwise CSF comparisons. These maps can be interpreted as follows: At each grid position $(x, y)$, filled entries indicate that metric values of CSF $y$ are ranked significantly better than those from CSF $x$ (across bootstrap samples), cross-marks indicate no significant superiority. Abbreviations: ncs: new-class shift (s for semantic, ns for non-semantic).

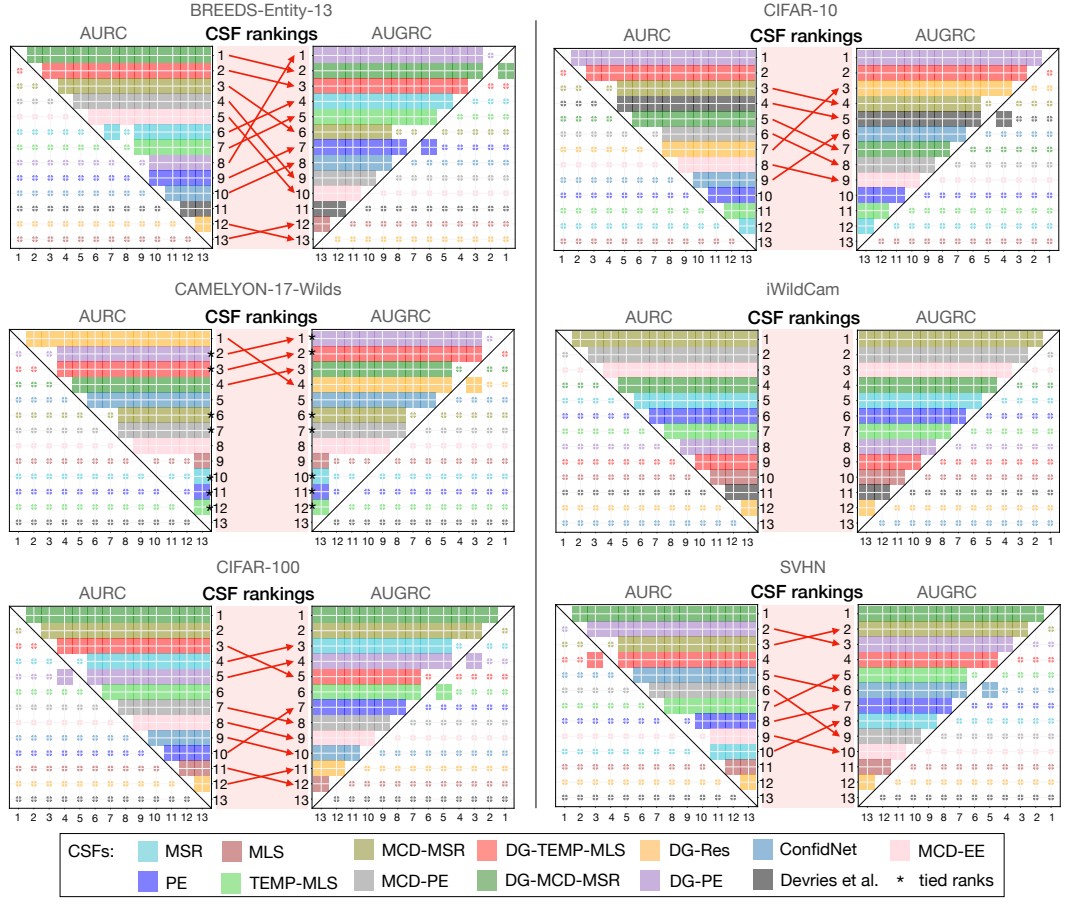

Figure 7: **Method ranking analysis without correction for multiple testing.** In contrast to the results shown in Figure 3, which include the Holm correction for multiple testing, this figure displays the results without correction. On 5 out of 6 datasets, the top-3 CSFs (out of 13 compared methods) change when employing the proposed AUGRC instead of AURC. This demonstrates the practical relevance of the AUGRC metric for Selective Classification evaluation. CSFs are color-coded and sorted from top (best) to bottom (worst) by average rank based on 500 bootstrap samples from the test dataset to ensure ranking stability. Ranking changes are reflected in changes in the color sequence and highlighted by red arrows. We assess the stability of the method rankings for each metric individually using one-sided Wilcoxon signed-rank tests, each with a 5% significance level, based on the bootstrap samples. Adjacent to each ranking, we present the resulting significance maps for the pairwise CSF comparisons. These maps can be interpreted as follows: At each grid position $(x, y)$, filled entries indicate that metric values of CSF $y$ are ranked significantly better than those from CSF $x$ (across bootstrap samples), cross-marks indicate no significant superiority. An ideal ranking exhibits only filled entries above the diagonal.

