# OpenReview forum: "Overcoming Common Flaws in the Evaluation of Selective Classification Systems"
_NeurIPS.cc/2024/Conference — NeurIPS 2024 spotlight_

### Official Review · Reviewer_QM33 · 2024-06-26

**Soundness:** 3
**Presentation:** 4
**Contribution:** 3
**Rating:** 7
**Confidence:** 3

**Summary:**

The authors introduce a new metric AUGRC for evaluating classifiers under the Selective Classification framework, whereas the classifier
has an option to reject low-confidence predictions. The authors introduce desirable properties for evaluating the Selection Classifiers, and show that the new metric has all those properties, unlike any of the currently used metrics. Experiments show that AUGRC produces significanly different rankings of Confidence Scoring Functions compared with currently used metrics.

**Strengths:**

The authors address relevant question and make useful contribution. The paper is well written, clear and easy to read. The experimental analyses are, to my judgement, sound. The code is provided.

**Weaknesses:**

Fig. 4: the risk vs. coverage curves for AURC vs. AUGRC do not look that different. Sure, there is non-monotonicity in AURC, but that is
in the low-coverage region, which presumably is not of practical interest. The rest looks monotonic and fairly similar. It is also true
that the two metrics suggest different CSFs, but again, they aren't that different. To be clear, I still think AUGRC is favorable, just
saying that the difference appears modest.

I think the key conclusion, which is the recommendation to adopt AUGRC, is valid, however the language in the section is a bit too strong (there is "substantial" twice and "significant limitations"). I recommend to tone-it-down.

**Questions:**

The authors claim that Selective Classification is a crucial components for clinical diagnostics ML systems (among other application domains). Can you provide examples of classifiers in clinical use that incorporate Selective Classification?

What is AUROC_f? It is introduced on line 172, however the description is unclear and there are no references.

**Limitations:**

There is no limitations section. I don't have a good idea regarding limitations of this manuscript.

---

> ### Author Rebuttal · Authors · 2024-08-06
>
> Thank you again for your valuable comments, and for taking the time to read our general reply, as well as considering our point-by-point comments here:
>
> ---
> W1. “Fig. 4: the risk vs. coverage curves for AURC vs. AUGRC do not look that different. Sure, there is non-monotonicity in AURC, but that is in the low-coverage region, which presumably is not of practical interest. The rest looks monotonic and fairly similar. It is also true that the two metrics suggest different CSFs, but again, they aren't that different. To be clear, I still think AUGRC is favorable, just saying that the difference appears modest.”
> * While the overall shape of the Selective Risk-Coverage is indeed similar to that of the Generalized Risk-Coverage curve, we demonstrate in our experiments that even smaller visual changes in the curves lead to alterations in method rankings and are thus highly relevant.
> * Further, we want to point out that the low-coverage region may also be relevant in practice, for instance considering a clinical deferral-system where a large fraction of cases may be deferred to ensure very high accuracy on the accepted cases. When evaluating specific applications for which certain risk or coverage intervals are known to be irrelevant, adaptations such as a partial AUGRC (analogous to the partial AUROC) may be considered. In response to your feedback, we added the note on partial evaluation of the AUGRC to the conclusion section.
>
> ---
> W2. “I think the key conclusion, which is the recommendation to adopt AUGRC, is valid, however the language in the section is a bit too strong (there is "substantial" twice and "significant limitations"). I recommend to tone-it-down.”
> * Thank you for this helpful comment. In the updated manuscript, we toned down the conclusion, as you proposed, removing the word “significant” in “the current metrics have significant limitations” (line 276) and “substantial” in “substantial deviations from intended and intuitive performance assessment behaviors” (line 282-283).
>
> ---
> Q1. “The authors claim that Selective Classification is a crucial components for clinical diagnostics ML systems (among other application domains). Can you provide examples of classifiers in clinical use that incorporate Selective Classification?”
> * Three interesting examples would be the following:
>   * Dvijotham et al., “Enhancing the reliability and accuracy of AI-enabled diagnosis via complementarity-driven deferral to clinicians”: The authors introduce the SC-based CoDoC deferral system for breast cancer screening and TB triaging, showing “that combined AI-clinician performance using CoDoC exceeds that currently possible through either AI or clinicians alone”.
>   * Leibig et al., “Combining the strengths of radiologists and AI for breast cancer screening: a retrospective analysis”: The authors propose an SC-based decision-referral approach for breast-cancer screening based on mammography data.
>   * Bungert et al., “Understanding Silent Failures in Medical Image Classification”: The authors comprehensively evaluate SC in the biomedical field and develop an interactive tool that facilitates identifying silent failures.
>
>
> ---
> Q2. “What is AUROC_f? It is introduced on line 172, however the description is unclear and there are no references.”
> * The “failure AUROC” AUROC$_f$ is the standard AUROC but computed on the binary failure labels (correctly classified vs. failure, denoted $Y_f$ in this paper) and the confidence scores. We added a corresponding reference in Section 2.4.
>
> ---
> Thank you for your constructive feedback. As we believe in having resolved your comments, please let us know in case there are remaining concerns that would prevent you from recommending a stronger acceptance of our work.

---

> > ### Comment · Reviewer_QM33 · 2024-08-12
> >
> > I appreciate your responses. I'll raise my rating to 7.

---

> > > ### Author Response · Authors · 2024-08-13
> > > **Thank you for your response**
> > >
> > > Thank you very much for taking the time to read our response and re-considering your assessment based on the provided updates.

---

### Official Review · Reviewer_izgv · 2024-06-27

**Soundness:** 2
**Presentation:** 2
**Contribution:** 3
**Rating:** 6
**Confidence:** 4

**Summary:**

The paper presents 5 requirements for multi-threshold metric for Selective Classification and a novel metric to evaluate selective classifiers called AUGRC. The proposed metric satisfies 5 requirements that are not met by current approaches. The proposed metric changes the rankings on 5 out of 6 datasets considered by the authors.

**Strengths:**

The paper's main strengths are:


1. the paper aims to tackle a long-standing concern in abstaining classifiers literature, i.e. how to evaluate with a single measure these classifiers
2. the theoretical derivations seem sound
3. the contribution is well framed within current literature

**Weaknesses:**

The main concerns of the paper are:

1. The empirical evaluation can be improved
2. The interpretability of the proposed measure is not that straightforward
3. The paper presentation can be improved.

Regarding the empirical evaluation, I have a few remarks.
* there are some contradictory lines, which should be double-checked: for instance, in lines 268-269, it is not clear to me why you claim that AURC erroneously favors DG-RES over MCD-PE, as DG-RES's performance seems to be better than MCD-PE (accuracy-wise and ranking-wise). Similarly, it is not clear to me why in Figure 4 the authors state DG-RES is favored "despite a lower classification performance and ranking quality", while in Figure 4.a is reported a higher accuracy and higher $AUROC_f$ for DG-RES;
* second, the authors never specify whether they correct for multiple outcomes testing (e.g. using Bonferroni correction). I think since the authors are performing multiple pairwise ranking tests, they also have to account for this.

Regarding the interpretability requirement, the authors claim (correctly) that the AUROC can be interpreted as "the probability of a positive sample having a higher score than a negative one". I did not fully grasp what is the straigthforward interpretation of the AUGRC score in this context (lines 206-208).

Regarding paper presentation, the paper heavily relies on acronyms. I would personally reconsider this choice to improve the overall readability.

**Questions:**

I have a few questions for the authors:

* [Q1] can the authors clarify my concerns regarding the empirical evaluation?
* [Q2] can the authors discuss further my concern regarding interpretability? Is there a straightforward interpretation of the AUGRC score?
* [Q3] when considering a degenerate abstaining classifier that always abstains, i.e., whose confidence score is always zero, is the AUGRC 0 (as the one obtained by a perfect classifier)? If so, is AUGRC favoring classifiers that tend to be underconfident and asbtain too much? Is there a way to avoid such a behaviour?
* [Q4] if I correctly understand how AUGRC works, why do not the authors consider the _classification quality_ by [Condessa et al, 2017] instead of the Generalized Risk Score? Such a metric also considers the failures that are correctly rejected and could avoid favoring abstaining classifiers that over reject.


Typos:

line 128: schore  $\to$ score

**Limitations:**

The authors do not directly discuss limitations of their proposed approach in the main paper. For instance, I think a brief discussion regarding the computational time required to compute AURGC should be included in the main paper.

---

> ### Author Rebuttal · Authors · 2024-08-06
>
> Thank you again for your valuable comments, and for taking the time to read our general reply, as well as considering our point-by-point comments here:
>
> ---
> W1. “The empirical evaluation can be improved.”
>
> W1.1. “there are some contradictory lines [...]”
> * Thank you for pointing out the mistakes in Section 4.2. It should read “erroneously favors MCD-PE over DG-RES” in line 268-269, and it should read “favoring MCD-PE [...] compared to DG-Res” in the caption of Figure 4. We corrected the sentences in the manuscript.
>
> W1.2. “the authors never specify whether they correct for multiple outcomes testing [...]”
> * Thank you for pointing this out. Please find our detailed response in the general response to all reviewers above.
>
> ---
> W2. “[...] straigthforward interpretation of the AUGRC score in this context (lines 206-208).”
> * Arguably most intuitively, the AUGRC corresponds to the rate of silent failures averaged over working points. In the suggested context, the interpretation is: On drawing two random samples, it corresponds to the probability that either both are failure cases or that one is a failure and has a higher confidence score than the non-failure. This can be read from Equation 7:
> $$AUGRC = (1 − AUROC_f ) · acc · (1 − acc) + (1 − acc)^2 / 2$$
>   The second term is the probability of drawing two failures. The first term is the probability that one is a failure times the probability that the failure is ranked higher (1 − AUROC$_f$). As an example, a CSF that outputs random scores yields AUROC$_f=0.5$, and hence AUGRC $=(1-acc)/2$.
> * In response to your feedback, we changed “expected risk of silent failures across working points” in lines 201-202 to “rate of silent failures averaged over working points”. Further, we extended the explanation on Equation 7 to include the AUROC-analogous interpretation as well as the example case of a random CSF.
>
> ---
> W3. “[...] the paper heavily relies on acronyms”
> * Thank you for raising this point. In response to your feedback, we reintroduced the abbreviations at important places in the paper (e.g. beginning of Section 2) or used the full term instead. We further updated Section 2.4 to introduce all acronyms with the full term for improved readability.
>
> ---
> W4. “[...] a brief discussion regarding the computational time required to compute AURGC should be included in the main paper.”
> * Thank you for bringing up this point. The main bottleneck of the AUGRC computation is the sort operation on the confidence scores (O(n log n)). This is the same as for the AURC and there is no computational overhead as compared to the AURC.
> * Benchmarking the computational time for evaluating the AURC, AUGRC, and AUROC$_f$ for 1k random scores and failure labels, we obtain the following results:\
> AUROC$_f$: 730μs ± 13μs (sklearn implementation)\
> AURC: 564μs ± 2μs (ours)\
> AUGRC: 562μs ± 2μs (ours)
> * In response to your feedback, we added a note on the AURC computation time in the beginning of Section 4, and details and benchmark results to a new section in the Appendix: “A.2.3 Computation Time”.
>
> ---
> Q1. & Q2.
> Addressed in W1-W3.
>
> ---
> Q3. “[...] confidence score is always zero, is the AUGRC 0 [...]? If so, is AUGRC favoring classifiers that tend to be underconfident and asbtain too much? [...]”
> * To clarify, whether a SC model abstains given a predicted confidence score depends on a threshold which is fixed on application. However, a multi-threshold evaluation, like in AUGRC, does not involve a fixed threshold but aggregates across all working points. More precisely, no explicit abstention decision is made in the AUGRC, the same as no classification decision is made in the classification AUROC.
> * Illustrating the latter with an example: A binary classifier that always outputs $P(Y=1)=0$ will yield an AUROC of ½ being equivalent to the random classifier. Analogously, if the confidence scores are always zero, the AUROC$_f$ is ½ and following Equation 7 the AUGRC is $(1-acc)/2$. Thus, the AUGRC does not favor underconfident classifiers.
> * We extended the explanation on Equation 7 (see our response to W2).
>
> ---
> Q4. “[...] why do not the authors consider the classification quality [...]? [...]”
> * [2] define the Classification quality (Q) as the fraction of rejected failures plus the fraction of non-rejected non-failures.
> * Using 1-Q as a replacement of Generalized Risk in the AUGRC deviates from the intention of evaluating the silent failure risk, as this assigns a loss of 1 to rejected but correctly classified samples. As an example of resulting unintended behavior, two classifiers with accuracy 0 and 1, respectively, both yield a Q-based AUGRC of ½. This breaks requirement R2.
> * Please note that instead of a holistic evaluation suitable for method benchmarking, [2] derives “performance measures [...] [that] correspond to a reference operating point”. Thus, Q may be used for working point selection but is not suitable for replacing Generalized Risk in the AUGRC.
> * We added to Section 2.1: “Aside from the selective risk, other performance measurements for working point selection include for example Classification quality and Rejection quality [2]”.
>
> [2] Condessa et al., “Performance measures for classification systems with rejection”
>
> ---
> Thank you for your constructive feedback. As we believe to have resolved your comments, please let us know if there are any remaining concerns that would hinder a recommendation for acceptance.

---

> > ### Comment · Reviewer_izgv · 2024-08-08
> > **Response to the Rebuttal**
> >
> > I thank the authors for their clarifications and the new experiments.
> > A few comments:
> >
> > regarding Q1, I thank the reviewers for implementing my suggested changes.
> >
> > regarding Q2,
> > > the interpretation is: On drawing two random samples, it corresponds to the probability that either both are failure cases or that one is a failure and has a higher confidence score than the non-failure. This can be read from Equation 7:
> >
> > I think clarifying the interpretation of AUGRC (meaning adding these lines to the paper) is very important and can help the reader improving its understanding.
> >
> > Regarding Q3 and Q4, I see your points, and I have no further questions.
> >
> > Finally, I thank the authors for providing a small analysis regarding the required time for computing AUGRC.
> >
> > Hence, after the authors' rebuttal and clarifications, I will increase my score to a weak accept.

---

> > > ### Author Response · Authors · 2024-08-13
> > > **Thank you for your response**
> > >
> > > Thank you very much for taking the time to read our response and re-considering your assessment based on the provided updates.

---

### Official Review · Reviewer_xFt7 · 2024-07-06

**Soundness:** 4
**Presentation:** 4
**Contribution:** 4
**Rating:** 8
**Confidence:** 5

**Summary:**

The authors propose 5 requirements that should be satisfied by selective classification (SC) metrics such that they can be successfully used to rank SC models for a task. They then propose a new metric, called AUGRC, which is shown to satisfy all 5 requirements. Finally, the authors show empirically that their method performs better rankings thank previous metrics.

**Strengths:**

1. The paper is well-motivated, clearly written, and has an excellent coverage of previous works.
2. The 5 requirements were well-chosen and a metric that satisfies them is likely to be a good choice for SC ranking.
3. The metric is simple, yet effective, and the maths seems sound.
4. Results are convincing and statistically analysed; the toy dataset was didactic.
5. There's enough information in the main body of the paper to fully understand the experiments and corresponding results, i.e. it's not necessary to read the appendix, even if further details are available there.

**Weaknesses:**

1. Limited discussion of future works.
2. NLL and Brier Score are listed among the multi-threshold metrics, but there's no thresholding involved in their evaluation. In fact, it should be possible to use them as risk measures, which would cover more for than just 0/1 loss, especially where probabilities are important for decision making.

**Questions:**

1. A similar task to SC is classification with rejection of OOD samples (sometimes called open-set recognition). In such cases, there isn't a ground truth to evaluate the risk on (in practice). Can the authors envision a way to adapt their metric to these scenarios?
2. The metrics were scaled by ×1000 in Figure 2. Were they also rescaled for all the other results?
3. Regarding interpretability, the authors say that "The optimal AUGRC is given by the second term in Equation 7". What does the optimal AUGRC mean? What would be the AUGRC of the Bayes-optimal model for a task? Is there an AUGRC value associated to a random classifier, as in AUROC?
5. Any intuitions about why the rankings of AURC and AUGRC where the same for iWildCam?

**Limitations:**

Yes.

---

> ### Author Rebuttal · Authors · 2024-08-06
>
> Thank you again for your valuable comments, and for taking the time to read our general reply, as well as considering our point-by-point comments here:
>
> ---
> W1. “Limited discussion of future works.”
> * Thank you for this helpful comment. In response to your feedback, we extended the Conclusion Section to point out directions of future work: Our proposed evaluation framework provides a solid basis for future work in Selective Classification, including developing novel methods as well as analyzing the properties of individual methods.
>
> ---
> W2. “NLL and Brier Score are listed among the multi-threshold metrics, but there's no thresholding involved in their evaluation. In fact, it should be possible to use them as risk measures, which would cover more for than just 0/1 loss, especially where probabilities are important for decision making.”
> * Thank you for raising this point. Indeed, the NLL and Brier Score do not involve thresholding of confidence scores. In response to your feedback, we rephrased line 167 to: “Importantly, proper scoring rules such as the Negative-Log-Likelihood and the Brier Score are technically not multi-threshold metrics. Yet, we include them here as they also aim for a holistic performance assessment, i.e. assessment beyond individual working points.”
> * Regarding your proposal to use NLL or Brier Score as risk measure: NLL and BS both assess the calibration of confidence scores in addition to their ranking. Generally, the calibration task can be viewed as orthogonal to that of Selective Classification where rejection is solely based on the ranking of scores. For a detailed discussion on the relation between calibration and confidence ranking we refer to Appendix C in [1]. Further investigating the relation between confidence ranking and calibration and on evaluating SC in settings where calibrated scores are of interest is an interesting direction of future work, which we added to the conclusion section in the updated manuscript.
>
> [1] Jäger et al., “A Call to Reflect on Evaluation Practices for Failure Detection in Image Classification”
>
> ---
> Q1. “A similar task to SC is classification with rejection of OOD samples (sometimes called open-set recognition). In such cases, there isn't a ground truth to evaluate the risk on (in practice). Can the authors envision a way to adapt their metric to these scenarios?”
> * We understand you are referring to settings where the SC model is validated on data containing classes that were not part of the training data. The AUGRC metric based on the binary failure label is directly applicable to these scenarios, as any prediction on unknown classes is per definition a misclassification (for a visualization, see Figure 5 in [1]). In Table 3, we report AUGRC values evaluated on various distribution shifts, also containing semantic and non-semantic new-class shifts.
> * In case you are referring to a different kind of open-set recognition, we would ask for  a more concrete task formulation for us to comment on.
>
> ---
> Q2. “The metrics were scaled by ×1000 in Figure 2. Were they also rescaled for all the other results?”
> * Thank you for pointing out the missing information. We rescaled all results apart from Figure 5. More precisely, we scaled the AURC and AUGRC values by 1000 in Figure 4 and Table 3. For the color-coded AUGRC in Figure 5, no rescaling was done. In response to your feedback, we updated the caption of Figure 4 to clearly state which values were rescaled.
>
> ---
> Q3. “What does the optimal AUGRC mean? What would be the AUGRC of the Bayes-optimal model for a task? Is there an AUGRC value associated to a random classifier, as in AUROC?”
> * The optimal AUGRC refers to the AUGRC for an optimal CSF, i.e. one that ranks all failure cases lower than correctly classified cases (AUROC$_f = 1$), but is conditioned on the performance of the underlying classifier. Please note that the CSF is not directly linked to the classifier output, but can be any (external) function that provides continuous confidence scores per classified sample. As such, in the general formulation, classifier and CSF are two independent entities and AUGRC assesses the performance of both jointly.
> * The Bayes-optimal classifier by itself does not directly define a CSF and hence does not correspond to a specific AUGRC value. We believe that investigating the relation between Bayes-optimality and SC is an interesting direction of future work.
> * On the level of detecting failure cases, i.e. given a fixed classifier with accuracy $acc$, assigning random confidence scores yields a AUROC$_f=0.5$. Following Equation 8, this corresponds to AUGRC $=(1-acc)/2$.
> The same holds if the classification itself is random. There, however, the accuracy depends on the number of classes $K$, i.e. $acc=1/K$.
> * In response to your questions, we re-formulated line 206 to: “The AUGRC for an optimal CSF (AUROC$_f=1$) is given by the second term in Equation 7”. We also added the AUGRC of the random classifier to the same section.
>
> ---
> Q4. “Any intuitions about why the rankings of AURC and AUGRC where the same for iWildCam?”
> * For the iWildCam dataset, we observe very robust method rankings for both AURC and AUGRC which don’t change between the two metrics. Overall, the volatility of method rankings varies from dataset to dataset (Figure 3).
> * We agree that the investigation of specific methods and datasets is an interesting direction of future work, which we added to the conclusion section in the updated manuscript.
>
> ---
> Thank you once more for your constructive feedback. As we believe to have resolved your comments, please let us know in case you have remaining suggestions to further increase the quality of our work.

---

### Official Review · Reviewer_BBe8 · 2024-07-16

**Soundness:** 3
**Presentation:** 3
**Contribution:** 3
**Rating:** 7
**Confidence:** 4

**Summary:**

The paper tackles the problem of Selective Classification (SC). The authors show problem with the existing metrics used in evaluating SC and propose to use the Area under the Generalized Risk Coverage curve (AUGRC). Empirical results provide useful insights about the effect of using this metric and shows how this changes  the relative ordering of state-of-the art methods.

**Strengths:**

- The problem tacked is very important, especially for the safe deployment of ML models which requires the knowledge of when not to trust the model.
- The proposed metric is well-motivated.
- The experiments are extensive.
- It is important for the community to know about this work as this changes the perception of SC and the best method to use.

**Weaknesses:**

- It is important to add the details of the methods in the main paper. For example, DeepGamblers (DG) is referred to as DG in the main paper and this abbreviation is only explained in the appendix. Similarly for other baselines.
- Although the experiments are extensive, it is good to include the recent state-of the art SC methods such as Feng et al (2023) to understand which method is the best to use.

References:
Feng et al: Towards Better Selective Classification

**Questions:**

See the previous section

---

> ### Author Rebuttal · Authors · 2024-08-06
>
> Thank you again for your valuable comments, and for taking the time to read our general reply, as well as considering our point-by-point comments here:
>
> ---
> W1. “It is important to add the details of the methods in the main paper. For example, DeepGamblers (DG) is referred to as DG in the main paper and this abbreviation is only explained in the appendix. Similarly for other baselines.”
> * Thank you for pointing out that the abbreviations DG and MCD-PE were used in the main text without reference to the explanation in the appendix. This was particularly the case in Section 4.2, where we explicitly compare these two confidence scoring functions. In response to your feedback, we adapted Section 4.2 to properly introduce the abbreviations DG and MCD-PE.
> * While the main paper is now fully self-contained and all abbreviations are introduced, we would like to comment on the general decision to provide the details of utilized confidence scoring functions in the appendix. This is because the focus of our work is not on the confidence methods like in other studies, but on the metric, which is why we find it important to allocate sufficient space in the main paper to metric-related descriptions.
>
> ---
> W2. “Although the experiments are extensive, it is good to include the recent state-of the art SC methods such as Feng et al (2023) to understand which method is the best to use.”
> * Please note that the focus of our experiments is to show the relevance of the proposed metric rather than the performance of individual methods. We argue that this metric relevance can be demonstrated with a diverse and representative set of prevalent confidence scoring functions, but it is not necessary to include all existing methods.
> * The main conclusion in [2] is that “[...] selecting according to classification scores is the SOTA selection mechanism for comparison” (based on experiments with three SC methods, including DG). We want to point out that with DG-MCD-MSR, DG-PE, and DG-TEMP-MLS, our empirical evaluation already includes multiple CSFs that are trained with DG loss attenuation but are based on the classifier’s logits, aligning with the recommendation in [2].
> * We agree that testing and comparing further novel methods with our proposed AUGRC metric is an interesting direction for future work. However, running all experiments required for this method within one week is not feasible for us.
> In response to your feedback, we extended the first paragraph in Section 4 to include the method of Feng et al. in the discussion about future work.
>
> [2] Feng et al., “Towards better Selective Classification”
>
> ---
> Thank you once more for your constructive feedback. As we believe to have resolved your comments, please let us know in case you have remaining suggestions to further increase the quality of our work.

---

### Author Rebuttal · Authors · 2024-08-06

We sincerely thank all reviewers for their valuable comments. The reviewers generally agreed on the added value of our work , noting that “The paper is well written, clear and easy to read” (QM33) , “The metric is simple, yet effective” (xFt7), “The experiments are extensive” (BBe8), and “It is important for the community to know about this work as this changes the perception of SC and the best method to use” (BBe8).

However, one reviewer (izgv) did not yet recommend acceptance of the paper. Thus, additionally to the point-by-point responses below, we would like to address izgv’s main point of criticism here:

---
**Empirical evaluation: Missing correction for multiple testing in Section 4.1.** Reviewer izgv noted that a correction for multiple testing may be required when performing multiple pairwise ranking tests.
* We thank reviewer izgv for bringing up this point. We agree that when deducing overall stability of method rankings based on multiple pairwise tests, a correction for multiple testing such as the Bonferroni [1] or Holm [2] method is necessary.
* To address this concern, we conducted the empirical evaluation with both the Holm correction and the more conservative Bonferroni correction for multiple testing.
* Please find the detailed **results in the additional PDF**. Across the whole study, 913/936 pairwise differences were significant at alpha=0.05 without correction, 906/936 with the Holm correction, and 906/936 with the Bonferroni correction. In summary, the corrections did not alter our original statement that the results of the pairwise signed-rank tests “indicate stable method rankings for both AURC and AUGRC”.
* In response to the feedback we updated Figure 3 in the manuscript to include the Holm correction for multiple testing, following [3]. Further, we added the results without correction and with the Bonferroni correction to the appendix in section A.3 Additional Results.

[1] Dunn, “Multiple comparisons among means”\
[2] Holm, “A simple sequentially rejective multiple test procedure”\
[3] Wiesenfarth et al., “Methods and open‑source toolkit for analyzing and visualizing challenge results”

---
We have thoroughly revised our manuscript to address the provided feedback. The list of changes we performed  includes:
* We improved the explanations on the interpretability of the AUGRC in Section 3 based on the comments and questions by the reviewers xFt7 and izgv.
* We corrected two mistakes in the main text in Section 4.2, pointed out by izgv.
* We updated the significance maps displayed in Figure 3 to include the Holm correction for multiple testing. (izgv)
* We toned down strong formulations in the conclusion. (QM33)
* We renamed “multi-threshold metrics” to “holistic metrics” in several parts of the manuscript. (xFt7)
* We extended the Conclusion Section to point out directions of future work. (xFt7)
* We added details on the AUGRC computation time in the Appendix and Section 4. (izgv)
* We reduced the amount of acronyms used to improve clarity and readability. (izgv & BBe8)

---
We believe that these updates resolve the stated concerns of all reviewers. Please find our point-by-point answers in the respective reviewer sections.

---

### Comment · Area_Chair_xygR · 2024-08-12
**Author-Reviewer Discussion Phase**

Dear Authors and Reviewers!

Thank you for your reviews, rebuttal, additional comments, questions and responses!

We have the last two days of the discussion phase! Please use this time as efficiently as possible :)

Thank you,

NeurIPS 2024 Area Chair

---

### Decision · Program_Chairs · 2024-09-25

**Decision:**

Accept (spotlight)

**Comment:**

The paper addresses the problem of performance measurement in selective classification (SC). The authors introduce five key requirements that SC metrics should meet and demonstrate that none of the existing metrics fully satisfies all of these requirements. They propose a new metric, called the Area under the Generalized Risk Coverage curve (AUGRC), and show that it meets all five criteria. Additionally, empirical studies provide further evidence that this new metric can be more effective in evaluating SC algorithms.

All reviewers agree that the paper is worth publishing. In their rebuttal and during the discussion phase, the authors successfully addressed the reviewers' concerns and committed to making the necessary revisions.